# The barley pan-genome reveals the hidden legacy of mutation breeding

Murukarthick Jayakodi[1,20], Sudharsan Padmarasu[1,20], Georg Haberer[2],
Venkata Suresh Bonthala[2], Heidrun Gundlach[2], Cécile Monat[1], Thomas Lux[2], Nadia Kamal[2],
Daniel Lang[2], Axel Himmelbach[1], Jennifer Ens[3], Xiao-Qi Zhang[4], Tefera T. Angessa[4],
Gaofeng Zhou[4,5], Cong Tan[4], Camilla Hill[4], Penghao Wang[4], Miriam Schreiber[6],
Lori B. Boston[7], Christopher Plott[7], Jerry Jenkins[7], Yu Guo[1], Anne Fiebig[1], Hikmet Budak[8],
Dongdong Xu[9], Jing Zhang[9], Chunchao Wang[9], Jane Grimwood[7], Jeremy Schmutz[7],
Ganggang Guo[9], Guoping Zhang[10], Keiichi Mochida[11,12,13], Takashi Hirayama[13], Kazuhiro Sato[13],
Kenneth J. Chalmers[14], Peter Langridge[14], Robbie Waugh[6,14,15], Curtis J. Pozniak[3], Uwe Scholz[1],
Klaus F. X. Mayer[2,16], Manuel Spannagl[2], Chengdao Li[4,5,17 ✉], Martin Mascher[1,18 ✉] & Nils Stein[1,19 ✉]

Genetic diversity is key to crop improvement. Owing to pervasive genomic structural
variation, a single reference genome assembly cannot capture the full complement of
sequence diversity of a crop species (known as the 'pan-genome'[1]). Multiple
high-quality sequence assemblies are an indispensable component of a pan-genome
infrastructure. Barley (*Hordeum vulgare* L.) is an important cereal crop with a long
history of cultivation that is adapted to a wide range of agro-climatic conditions[2]. Here
we report the construction of chromosome-scale sequence assemblies for the
genotypes of 20 varieties of barley—comprising landraces, cultivars and a wild
barley—that were selected as representatives of global barley diversity. We catalogued
genomic presence/absence variants and explored the use of structural variants for
quantitative genetic analysis through whole-genome shotgun sequencing of
300 gene bank accessions. We discovered abundant large inversion polymorphisms
and analysed in detail two inversions that are frequently found in current elite barley
germplasm; one is probably the product of mutation breeding and the other is tightly
linked to a locus that is involved in the expansion of geographical range. This
first-generation barley pan-genome makes previously hidden genetic variation
accessible to genetic studies and breeding.

A staple food of ancient civilizations, today barley is used mainly for animal feed and malting. Barley is more adaptable to harsh environmental conditions than its close relative wheat, and maintains an important role in human nutrition in harsh climatic regions that include the Ethiopian and Tibetan highlands[2]. As in other crops, genomics has been a major driver of progress in barley genetics and breeding in the past decade[3]. The first draft reference genome for barley[4], and its subsequent revisions[5,6], have formed the basis for gene isolation[7], compiling a single-nucleotide polymorphism (SNP) variation atlas for wild and domesticated germplasm[8], and activating plant genetic resources[9]. At the same time, reduced-representation surveys of structural variation[10] and map-based cloning[11] have implicated variation

in gene content and copy number in the control of agronomic traits. The concept of the pan-genome refers to a species-wide catalogue of genic presence/absence variation (PAV)[12], or more generally, structural variation that affects (potentially non-coding) sequences of 50 or more base pairs (bp) in size. Although several methods of pan-genomic analysis that use short-read sequence data in the context of a single reference genome have been devised[13], large and complex genomes require multiple high-quality sequence assemblies to capture and contextualize sequences that are absent in—or highly diverged from—a single reference genotype[14]. Progress in sequencing and genome mapping technologies has only recently made possible the fast and cost-effective assembly of tens of genotypes of

[1]Leibniz Institute of Plant Genetics and Crop Plant Research (IPK) Gatersleben, Seeland, Germany. [2]Plant Genome and Systems Biology (PGSB), Helmholtz Center Munich, German Research Center for Environmental Health, Neuherberg, Germany. [3]Department of Plant Sciences, University of Saskatchewan, Saskatoon, Saskatchewan, Canada. [4]Western Barley Genetics Alliance, State Agricultural Biotechnology Centre, College of Science, Health, Engineering and Education, Murdoch University, Murdoch, Western Australia, Australia. [5]Agriculture and Food, Department of Primary Industries and Regional Development, South Perth, Western Australia, Australia. [6]The James Hutton Institute, Dundee, UK. [7]HudsonAlpha, Institute for Biotechnology, Huntsville, AL, USA. [8]Montana BioAg Inc, Missoula, MT, USA. [9]Institute of Crop Sciences, Chinese Academy of Agricultural Sciences (ICS-CAAS), Beijing, China. [10]College of Agriculture and Biotechnology, Zhejiang University, Hangzhou, China. [11]Bioproductivity Informatics Research Team, RIKEN Center for Sustainable Resource Science, Yokohama, Japan. [12]Kihara Institute for Biological Research, Yokohama City University, Yokohama, Japan. [13]Institute of Plant Science and Resources, Okayama University, Kurashiki, Japan. [14]School of Agriculture, Food and Wine, University of Adelaide, Glen Osmond, South Australia, Australia. [15]School of Life Sciences, University of Dundee, Dundee, UK. [16]School of Life Sciences Weihenstephan, Technical University of Munich, Freising, Germany. [17]Hubei Collaborative Innovation Centre for Grain Industry, Yangtze University, Jingzhou, China. [18]German Centre for Integrative Biodiversity Research (iDiv) Halle-Jena-Leipzig, Leipzig, Germany. [19]Center for Integrated Breeding Research (CiBreed), Georg-August-University Göttingen, Göttingen, Germany. [20]These authors contributed equally: Murukarthick Jayakodi, Sudharsan Padmarasu. ✉e-mail: C.Li@murdoch.edu.au; mascher@ipk-gatersleben.de; stein@ipk-gatersleben.de

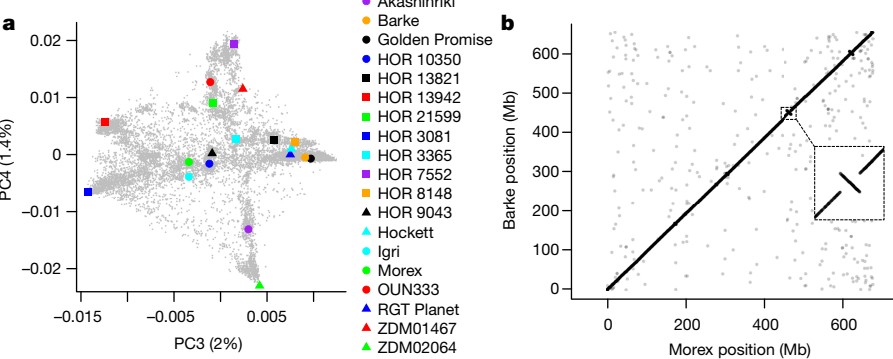

**Fig. 1 | Chromosome-scale sequences of 20 representative barley genotypes reveal large structural variants. a**, We selected 20 barley genotypes to represent the genetic diversity space, as revealed by PCA of genotyping-by-sequencing data of 19,778 domesticated varieties of barley[9]. Principal component (PC)3 and PC4 are shown. The proportion of variance explained by the principal components is indicated in the axis labels. Further

principal components are shown in Extended Data Fig. 1a. **b**, Alignment of the pseudomolecules of chromosome 2H of the Morex and Barke cultivars. The inset zooms in on a 10-Mb inversion that is frequently found in germplasm from northern Europe. Co-linearity plots for all assemblies and chromosomes are shown in Extended Data Fig. 3a.

large-genome plant species, such as barley (haploid genome size of 5 Gb)[15].

## Twenty barley reference genomes

The starting point for pan-genomics in barley was the comprehensive survey of species-wide diversity on the basis of the genome-wide genotyping of more than 22,000 barley accessions, mainly from the German national gene bank[9]. To achieve a good representation of major barley gene pools, we selected accessions that were located in the branches of the first six principal components from the previously published principal component analysis (PCA)[9] (Fig. 1a, Extended Data Fig. 1), reflecting the key determinants of population structure: geographical origin, row type and annual growth habit. In addition to these gene pool representatives, our panel included the reference cultivar Morex[5], two current or former elite malting varieties (RGT Planet and Hockett), two founder lines of Chinese barley breeding (ZDM01467 and ZDM02064), Golden Promise and Igri (two genotypes with high transformation efficiency[16,17]), Barke (a successful German variety and the parent of several mutant and mapping populations[18,19]) and one wild barley (*H. vulgare* subsp. *spontaneum* (K. Koch) Thell.) genotype from Israel (B1K-04-12, a desert ecotype collected at Ein Prat)[20].

We constructed chromosome-scale sequence assemblies for 20 accessions (Extended Data Table 1). In brief, paired-end and mate-pair Illumina short reads were assembled into scaffolds of megabase (Mb)-scale contiguity (Extended Data Table 1). Scaffold assembly was done with Minia[21] and SOAPdenovo[22] following the TRITEX method[6] (*n* = 16), DeNovoMagic from NRGene (*n* = 3) or W2rap[23] (*n* = 1). We used 10X Genomics Chromium linked-reads and chromosome conformation capture (Hi-C) data to arrange scaffolds into chromosomal pseudomolecules using the TRITEX pipeline[6] (Extended Data Table 1). A comparison of the short-read assembly of the Morex cultivar to a long-read assembly of this genotype generated from PacBio long reads showed high co-linearity at chromosomal scale, good concordance in gene space representation and similar power to detect PAV (Extended Data Fig. 2), indicating that short-read assemblies are amenable to pan-genomic analyses in barley. Although the assemblies of the 20 diverse accessions differed in contiguity and the extent of gap sequence in the intergenic space, they had a similar representation of reference gene models (Morex V2) and were highly co-linear to each other at the whole-chromosome scale (Fig. 1b, Extended Data Fig. 3). A similar proportion (about 80%) of the assembled sequence of each genotype was composed of transposable elements, with an average of 113,200 intact full-length long-terminal repeat retro-elements (LTRs)

per assembly (Supplementary Table 1). However, we found pronounced differences in the number of shared intact full-length LTR locations: only 17 to 25% of full-length LTR locations present in the wild barley B1K-04-12 were shared at 98% sequence identity and 98% alignment coverage with any domesticated genotype (Extended Data Fig. 4). By contrast, more closely related domesticated genotypes shared between 53% and 67% of their full-length LTRs, consistent with previous reports of rapid sequence turn-over in the non-coding space in large-genome plant species[24,25].

De novo gene annotation using Illumina RNA sequencing and PacBio Iso-Seq data (Supplementary Table 2) was performed for three genotypes: Morex (which has previously been reported[6]), Barke and the Ethiopian landrace HOR 10350 (Extended Data Fig. 5). Gene models defined on the basis of these three assemblies were consolidated and projected onto the remaining 17 assemblies (Extended Data Fig. 5). Between 35,859 and 40,044 gene models were annotated by projection in each assembly (Extended Data Table 1) with an average of 37,515 (s.d. = 896). The number of gene models was about 20% higher in the projections than in de novo annotations (Extended Data Fig. 5e), which indicates that some of the models lack transcript support: possible explanations for the discrepancy are highly tissue-specific expression or pseudogenization. The clustering of orthologous gene models yielded 40,176 orthologous groups. Of these, 21,992 occurred as a single copy in all 20 assemblies; 3,236 occurred in multiple copies in at least one of the 20 assemblies; 13,188 were absent from at least one assembly; and 1,760 were present in only one assembly. On average, 14.7% of gene models annotated in each assembly occurred in tandem arrays that comprised two or more adjacent copies. These results point to abundant genic copy-number variation between barley genotypes. Future transcriptomic studies will ascertain the effect of structural variants on gene expression.

## Pan-genome as a tool for genetics and breeding

High-quality genome assemblies are a resource for ascertaining and providing context to structural variants, which can then be genotyped in a wider set of germplasm using low-coverage or reduced-representation sequence data. We used two complementary approaches to detect structural variation: assembly comparison and clustering of single-copy sequences to derive markers that can be scored in short-read data. We used the Assemblytics[26] software to discover PAV by pair-wise comparison of 19 chromosome-scale assemblies to the Morex reference. We identified 1,586,262 PAVs, ranging in size from 50 to 999,568 bp, and observed an enrichment for low-frequency variants (Extended Data Fig. 6a, b). PAV density was higher in distal, gene-rich regions (Extended

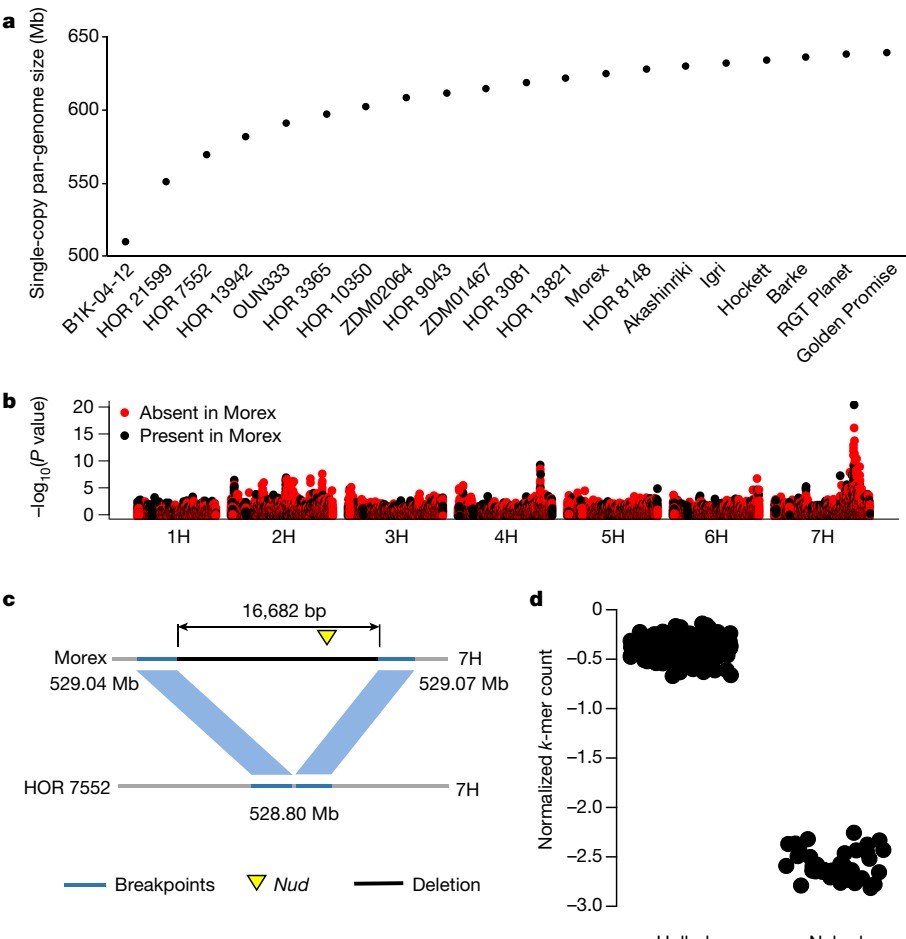

**Fig. 2 | Single-copy pan-genome and use of PAVs in association mapping.**
**a**, Cumulative size of single-copy regions in genome assemblies of 20 barley genotypes. The genotypes were ordered according to the size of their unique single-copy sequence. **b**, Genome-wide association scan for lemma adherence on the basis of PAV markers. The black and red dots in the Manhattan plot denote single-copy sequences that are present and absent in Morex, respectively. **c**, The most highly associated PAV marker was a 16.7-kb region that is deleted in the naked accession HOR 7552 and that contains the *NUD* gene[11]. **d**, Allelic status of the *NUD* deletion in 196 domesticated varieties of barley. Normalized single-copy *k*-mer counts within the 16.7-kb region are shown for hulled (*n* = 160 genotypes) and naked varieties (*n* = 36 genotypes).

Data Fig. 6c), which are characterized by higher nucleotide diversity and recombination rates[8]. A total of 5,446 out of 5,602 deletions longer than 5 kilobases (kb) found in Barke relative to Morex were mapped genetically in the 90 recombinant inbred lines of the Morex × Barke population[19] with highly concordant positions (Spearman correlation = 0.99) (Extended Data Fig. 6d), which provides support for the accuracy of the detected polymorphisms. At least one member of 18,562 (46%) groups of orthologous genes overlapped with structural variants discovered in the 20 sequence assemblies. As observed in other plant species[27], resistance-gene homologues containing NB-ARC and protein kinase domains were frequently found among PAV genes (Supplementary Table 3).

Structural variants cover non-genic regions composed of repetitive sequence, making it hard to establish orthologous relationships or the presence of specific alleles from short-read data only. To derive quantitative estimates of the extent of pan-genomic variation and as a tool for genetic analysis such as association scans, we focused on single-copy regions extracted from each of the 20 assemblies and clustered into a non-redundant set of sequences (hereafter referred to as the 'single-copy pan-genome') (Extended Data Fig. 7a). The average cumulative size of single-copy sequence in each accession was 478 Mb (that is, 9.5% of the assembly genome). The total size of non-redundant single-copy sequence was 638.6 Mb, represented by 1,472,508 clusters with an N50 of 1,087 bp (Extended Data Fig. 7b). The single-copy

sequence shared among all 20 genotypes amounted to 402.5 Mb, whereas 235.9 Mb were variable (that is, absent or present in higher copy number in at least one assembly) (Fig. 2a). On average, each of the 20 genotypes contained 2.9 Mb of single-copy sequence not present in any other assembly. As observed for transposable element divergence, the wild barley B1K-04-12 had the highest amount of unique single-copy sequence (Extended Data Table 1).

To test the suitability of the single-copy pan-genome for genetic analysis in a wider diversity panel without high-quality genome sequences, we collected whole-genome shotgun data (threefold coverage) for 200 domesticated and 100 wild varieties of barley (Supplementary Table 4). The abundance of 160,716 single-copy clusters that overlap structural variants was estimated by counting cluster-constituent *k*-mers (*k* = 31) in sequence reads of the diversity panel. In addition, we analysed genotyping-by-sequencing data of 19,778 gene bank accessions of domesticated barley[9] using the same approach. Abundance estimates based on *k*-mers (hereafter referred to as 'pan-genome markers') showed that loci detected as single-copy sequence in one genome assembly can vary in copy number from zero to many in diverse germplasm (Extended Data Fig. 7c). A PCA of pan-genome markers genotyped in whole-genome shotgun and genotyping-by-sequencing data highlighted the same drivers of global population structure as SNPs (Extended Data Fig. 7d–g). In genome-wide association scans for morphological traits, pan-genome markers revealed—with a good signal-to-noise ratio—peaks that are

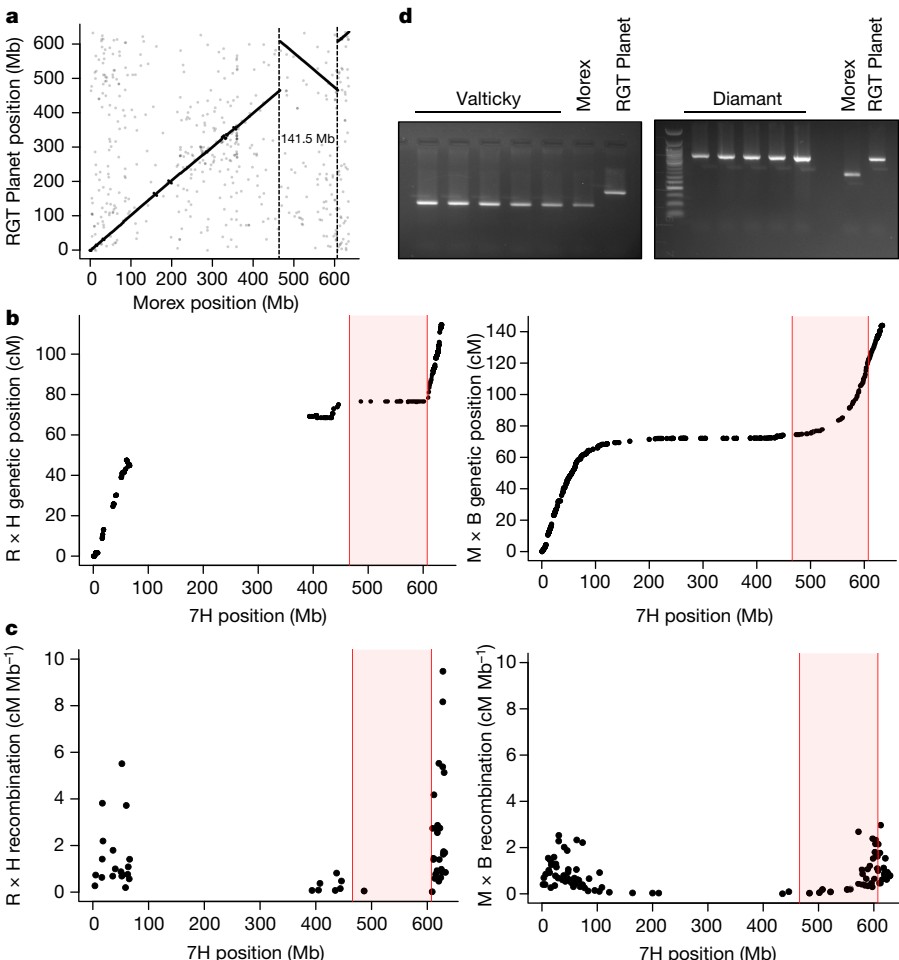

**Fig. 3 | Identification and characterization of a large inversion on chromosome 7H. a**, Alignment of the 7H pseudomolecules of the Morex and RGT Planet cultivars. **b**, Alignment of physical and genetic positions mapped in the RGT Planet × Hindmarsh (R × H) (left) and Morex × Barke (M × B) (right) populations. Red shading marks the inverted region. **c**, We converted genetic distances to recombination rates in the R × H (left) and M × B (right) populations. A single marker per recombination block is shown. **d**, We designed a PCR marker (Supplementary Figs. 1, 2a) to screen for the presence of the inversion in gene bank accessions that represent the Valticky and Diamant cultivars.

consistent with previous reports[9] (Fig. 2b, Extended Data Fig. 8). Notably, the pan-genome marker that was most highly associated with lemma adherence covered the *NUDUM* (*NUD*) gene[11] (Fig. 2c). All varieties of naked barley—in which lemmas can be easily separated from grains—are thought to trace back to a single mutational event, deleting the entire *NUD* sequence[11]. Another putative knockout allele of *NUD* (*nud1.g*) that contains a likely disruptive SNP variant was recently found in Tibetan barley[28]. All 36 naked accessions in our panel contained the known deletion (Fig. 2d), indicating that broader sampling of barley diversity—with a particular focus on centres of (morphological) diversity—is needed to discover novel rare alleles by genomic analyses.

Compared to reference-free approaches for *k*-mer-based genome-wide association scans such as AgRenSeq[29], trait-associated pan-genome markers are assigned with high precision to genomic positions, and aligning sequence assemblies in their vicinity provides immediate information about differences between haplotypes (Fig. 2c). Furthermore, the reduction of marker number by implicit clustering of *k*-mers into single-copy loci allows the use of standard mixed linear models[30,31] to correct for genomic relatedness.

## A map of polymorphic inversions

Chromosome-scale sequence assemblies can reveal large-scale rearrangements that are challenging to detect with other methods. Large inversions (more than 5 Mb in size) were prominent in the genome alignments of our 20 assemblies (Fig. 1b, Extended Data Fig. 3a, c). Previous reports on segregating inversions in barley are anecdotal and have focused on induced mutants[32,33]. To discover inversions in a broader set of germplasm, we mined patterns of contact frequencies in Hi-C data of a diversity panel mapped to a single reference genome[34]. Among 69 barley genotypes (67 domesticated and 2 wild accessions) (Supplementary Table 5), Hi-C-based inversion scans revealed a total of 42 events that ranged from 4 to 141 Mb in size (mean size of 23.9 Mb) (Extended Data Fig. 9a). Most of these events occurred in the low-recombining pericentromeric regions of the barley chromosomes and segregated at low frequency: 25 events were observed only once (Extended Data Fig. 9b, c, Supplementary Table 6). We focus here on two notable examples: a frequent event on chromosome 2H and an inversion in the distal region of the long arm of chromosome 7H.

The inversion in chromosome 7H detected in the RGT Planet cultivar was the largest event that segregated in our panel (141 Mb) (Fig. 3a). In a biparental mapping population derived from a cross between RGT Planet and the non-carrier cultivar Hindmarsh (Fig. 3b), this event repressed recombination in an interval that spanned 49 cM in the genetic map of the Morex × Barke population[19], which is isogenic for absence of the inversion (Fig. 3c, Supplementary Table 7). We also observed a moderately distorted segregation (57% allele frequency, $\chi^2 = 4.88$, $P < 0.05$) in favour of the Hindmarsh allele in this interval.

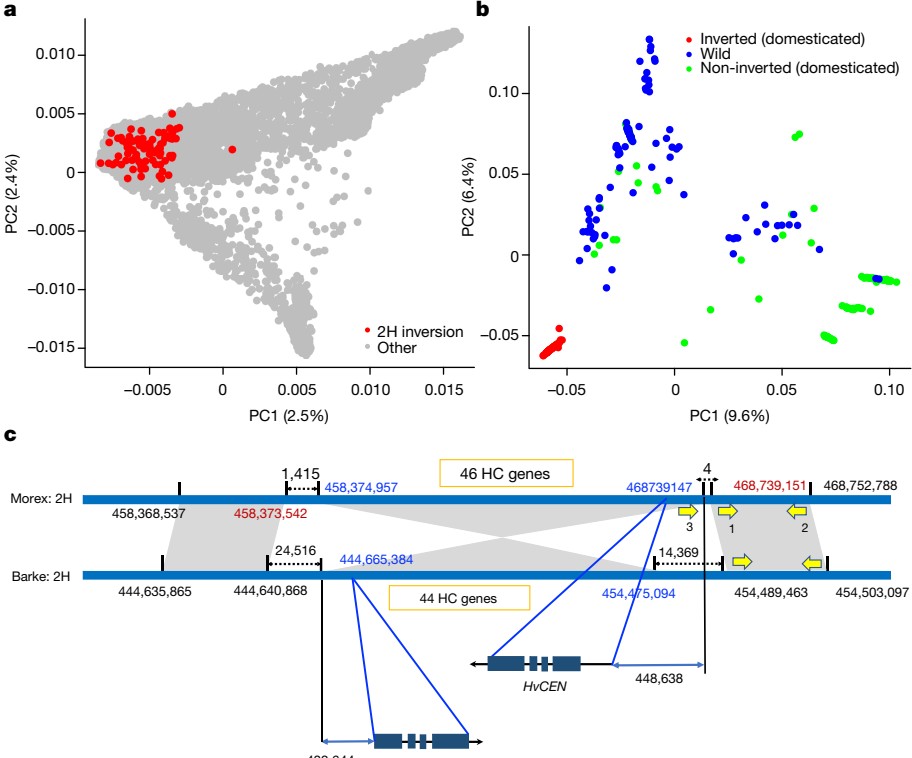

**Fig. 4 | Analysis of a frequent inversion on chromosome 2H. a**, A PCA showing the localization of inversion carriers in the diversity space of global domesticated barley. The correspondence of PCA coordinates to correlates of population structure is shown in Extended Data Fig. 1. Red dots denote carriers of the inverted haplotype (*n* = 87) in a panel of 200 domesticated varieties of barley. **b**, PCA for a diversity panel comprising 200 domesticated (red and green dots) and 100 wild varieties of barley (blue dots). SNP markers detected in whole-genome shotgun data and located in the inverted regions were used. **c**, Schematic of the inverted region. The *HvCEN* gene is closest to the breakpoint that is distal in Morex (distance of 449 kb) and proximal in Barke (distance of 433 kb) assemblies. A total of 46 and 44 high-confidence (HC) genes were annotated in the Morex and Barke assemblies, respectively. The yellow arrows (not drawn to scale) mark the positions of PCR primers to probe for the presence of the inversion (Supplementary Fig. 2c).

Recombination frequencies were increased in the flanking regions of the inversion in the RGT Planet × Hindmarsh population relative to Morex × Barke, which suggests a compensatory mechanism to maintain an average number of one-to-two crossovers per chromosome in the presence of large tracts of suppressed recombination[35].

By focusing on the inversion breakpoints in the RGT Planet sequence assembly, we designed a diagnostic PCR assay (Supplementary Fig. 2a, b, d) to rapidly genotype the presence of the inversion in 1,406 accessions (Supplementary Table 8). The inverted haplotype occurred at low frequency (1.3%) in the whole panel, but was found in many lines in the RGT Planet pedigree (Supplementary Fig. 3)—including commercially successful barley cultivars of past decades, such as Triumph, Quench and Sebastian. The earliest cultivar that carried the inversion was Diamant. As one of the donors of the semi-dwarf growth habit, Diamant was a highly influential founder line of modern barley breeding and traces back to a mutant induced by gamma irradiation of the Czech cultivar Valticky[36]. We genotyped several gene bank accessions and germplasm samples of both Valticky and Diamant. Notably, none of the Valticky samples carried the inversion, whereas it segregated in the Diamant samples (Fig. 3d). Quantitative trait loci mapping for yield-related traits in the RGT Planet × Hindmarsh population did not show signals on chromosome 7H (Supplementary Fig. 2e, Supplementary Table 9), consistent with selective neutrality of the inversion. This strongly suggests that mutation breeding in the 1960s has given rise to a cryptic large inversion, which—unbeknownst to breeders—segregates in elite varieties of barley.

The second inversion we focused on spanned 10 Mb in the interstitial region of chromosome 2H (Fig. 1b) and was present in 26 out of 69 Hi-C samples (Supplementary Table 8). Local PCA and haplotype analysis in our panel of 200 domesticated and 100 wild varieties of barley indicated a single origin of the inverted haplotype (Fig. 4a, b, Supplementary Fig. 2c). The inversion occurred only among domesticated barley of Western geographical origin[9], indicating that it arose or has risen to high frequency after domestication. The inverted region contains 46 high-confidence genes in the Morex cultivar. The closest gene to the inversion breakpoint—at 448 kb distance from the distal breakpoint in the non-carrier Morex—was *HvCENTRORADIALIS* (*HvCEN*)[37] (Fig. 4c). Although induced mutants of *HvCEN* flower very early, natural variation in *HvCEN* has previously been implicated in environmental adaptation to northern European climates[37]. All of the inversion carriers we analysed had *HvCEN* haplotype III, which is associated with later flowering in spring barley varieties from northern Europe[37,38]. Further research is required to determine whether the inversion close to *HvCEN* has direct functional consequences (for instance, by modulating *HvCEN* expression) or whether it hitchhiked along with a tightly linked causal variant.

## Discussion

The digital representation of the pan-genome can expand the repertoire of natural or induced sequence variation that is accessible to genetic analyses and breeding. Our comparison of 20 chromosome-scale sequence assemblies has revealed pervasive variation in genes and non-coding regions. Focusing on single-copy sequences, we translated this variation into scorable markers that are amenable to population genetic analysis and association scans. A notable finding was the prevalence of large (more than 5 Mb in size) inversion polymorphisms in current elite germplasm. It is likely that the suppression of genetic recombination in inversion heterozygotes has manifested

itself in hard-to-explain patterns of long-range linkage and segregation distortion between elite lines in breeding programmes. Our map of inversion polymorphisms will provide breeders with a point of reference to avoid—or interpret correctly—crosses between carriers and non-carriers. We found abundant structural variation in 20 representative barley genotypes, but individual events occurred at low frequency (Extended Data Figs. 6, 9). This observation, combined with the slow saturation of the single-copy pan-genome (Fig. 2a), motivate the genomic analysis of more genotypes to expand the barley pan-genome. The next phase of barley pan-genomics will focus on an augmented panel of domesticated and wild germplasm, working towards the long-term goal of high-quality genome sequences of all barley plant genetic resources as part of a biodigital resource centre[39,40].

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

## Methods

No statistical methods were used to predetermine sample size. The experiments were not randomized, and investigators were not blinded to allocation during experiments and outcome assessment.

### Library preparation, sequencing data generation and genome assembly of 20 diverse varieties of barley

High-molecular-weight DNA was extracted from one-week-old seedlings of 20 diverse barley accessions given in Supplementary Table 10, using a previously described large-scale DNA extraction protocol[41]. For the NRGene DeNovoMAGIC3.0 assemblies, 450-bp paired-end (PE450) libraries of Morex, Barke, HOR 10350 and B1K-04-12 were prepared at the Leibniz Institute of Plant Genetics and Crop Plant Research (IPK) Gatersleben. The 450-bp paired-end libraries for other accessions, 800-bp paired-end libraries and mate-pair libraries of three sizes were prepared and sequenced at the University of Illinois Roy J. Carver Biotechnology Center. The 10X Genomics Chromium libraries were prepared at the University of Saskatchewan Wheat Molecular Breeding Laboratory and sequenced by Genome Quebec or prepared and sequenced at the Roy J. Carver Biotechnology Center, using the manufacturers' recommendations. Published tethered chromosome conformation data for Morex, Barke, HOR 10350 and B1K-04-12 (ref. [42]) was used for scaffolding the respective genome. For the other accessions, in situ Hi-C libraries were prepared using a previously described method[43]. Sequencing data generated from each of the libraries are given in Supplementary Table 10. NRGene DeNovoMAGIC3.0 scaffold assemblies were provided for Barke, HOR 10350 and B1K-04-12. The 10X Chromium, population sequencing (POPSEQ) and Hi-C data were then used to prepare chromosome-scale assemblies using the TRITEX pipeline[6] (commit: 7041ff2). For the other assemblies, the TRITEX pipeline was also used for contig assembly and scaffolding with mate-pair and 10X data (Extended Data Table 1). High-confidence gene models annotated on the Morex V2 reference[6] and full-length cDNA sequences[44] were aligned to the assemblies to assess gene-space completeness with the parameters of ≥90% query coverage and ≥97% (≥90% for full-length cDNA) identity.

### Tissue collection and RNA extraction

Plant material for the collection of tissues for RNA sequencing (RNA-seq) and Iso-Seq was grown in the greenhouse at IPK Gatersleben with day–night temperatures of 21 °C–18 °C. Embryonic tissue, leaves, roots, internode, inflorescence (5 mm) and developing seeds (5 and 15 days after pollination) were collected as previously described[4], snap-frozen in liquid nitrogen and stored at −80 °C until RNA extractions were performed. RNA was extracted from the collected tissues using a Trizol extraction protocol[4] and purified using Qiagen RNeasy miniprep columns as per the manufacturer's instructions. RNA quality was checked on Agilent RNA HS screen tape and RNA with RIN value greater than 8 was used for RNA-seq and Iso-Seq library construction.

### RNA-seq library preparation and data generation

RNA-seq libraries were prepared from purified RNA using the TruSeq RNA sample preparation kit (Illumina) as per the manufacturer's recommendation at IPK Gatersleben. Libraries were pooled at equimolar concentrations, quantified by qPCR and paired-end-sequenced on an Illumina HiSeq 2500 for 200 cycles. The data generated for each tissue are given in Supplementary Table 2.

### Iso-Seq data generation and analysis

Two libraries for each embryonic tissue RNA and pooled RNA from seven tissues (described in 'Tissue collection and RNA extraction') were prepared for Barke and HOR 10350 using the PacBio Iso-Seq protocol. In brief, double-stranded RNA was synthesized using SMARTer PCR cDNA synthesis kit (Clontech; cat. no. 634925). Two fractions of cDNA with different size profiles were prepared by using differing ratios of DNA to Ampure XP beads (Beckman Coulter, cat. no. A63882). Equimolar concentration of each fraction were pooled, and a minimum of one microgram of double-stranded cDNA was used for Iso-Seq library construction as per the PacBio library construction protocol. Two additional libraries from pooled RNA tissues were prepared using cDNA prepared from TeloPrime v.1.0 kit (Lexogen) following the manufacturer's instructions. Libraries were quantified and sequenced on a PacBio Sequel device at IPK Gatersleben. Data were analysed using SMRTLink v.5.0 Isoseq v.1.0 pipeline or Isoseq3 pipeline (https://github.com/PacificBiosciences/IsoSeq_SA3nUP/wiki/Tutorial:-Installing-and-Running-Iso-Seq-3-using-Conda). The steps involved in Iso-Seq data analysis were the generation of circular consensus sequences, and then the classification of circular consensus sequence reads into full-length non-chimeric reads and non-full length reads on the basis of the presence of primer sequences and polyA sequences. Full-length non-chimeric reads were then clustered on the basis of sequence similarity to yield high- and low-quality isoforms. The data generated and method of library preparation are given in Supplementary Table 2.

### Gene projections and repeat annotation

Gene models for Morex, Barke and HOR 10350 were predicted using transcriptome data (Supplementary Table 2) and protein homology evidence, and derived by a previously described annotation pipeline[5]. High-confidence gene models from these accessions were aligned to pseudo-chromosomes of each accession separately using blat[45]. For each genomic region identified by blat, additional alignments were performed by exonerate[46] in its genomic neighbourhood ranging between 20 kb upstream and 20 kb downstream of the match position. A series of quality criteria was applied to select high-confidence gene models in each accession. The functional annotation for genes of 20 accessions was carried out using the AHRD pipeline v.3.3.3 (https://github.com/groupschoof/AHRD). Orthologous gene groups between the twenty accessions were predicted using OrthoFinder[47] v.2.3.1 with default parameters.

### Repeat annotation

To obtain a consistent transposon annotation across all lines for transposons and tandem repeats, the same methods were applied to all 20 barley lines. Transposons were detected and classified by homology search against the REdat_9.7_Poaceae section of the PGSB transposon library[48]. The program vmatch (http://www.vmatch.de, version 2.3.0) was used for that purpose as a fast and efficient matching tool that is well-suited for such large and highly repetitive genomes. Vmatch was run with the following parameters: identity ≥ 70%, minimal hit length 75 bp, seed length 12 bp (exact command line: -d -p -l 75 -identity 70 -seedlength 12 -exdrop 5). To remove overlapping annotations, the vmatch output was filtered for redundant hits via a priority-based approach. Higher scoring matches were assigned first and lower scoring hits at overlapping positions were either shortened or removed if they were contained to ≥90% in the overlap or <50 bp of rest length remained. The resulting transposon annotations are overlap-free, but disrupted elements from nested insertions have not been defragmented into one element. Still-intact full-length LTR retrotransposons were identified with LTRharvest[49], a program that scans the genome for LTR retrotransposon specific structural hallmarks, such as long terminal repeats, RNA cognate primer binding sites and target site duplications. LTRharvest (included in genometools 1.5.9) was run with the following parameter settings: 'overlaps best -seed 30 -minlenltr 100 -maxlenltr 2000 -mindistltr 3000 -maxdistltr 25000 -similar 85 -mintsd 4 -maxtsd 20 -motif tgca -motifmis 1 -vic 60 -xdrop 5 -mat 2 -mis -2 -ins -3 -del -3'. All candidates were annotated for PfamA domains using hmmer3 (http://hmmer.org, version 3.1b2) and filtered to remove false positives. The inner domain order served as a criterion for the LTR-retrotransposon superfamily classification into either Gypsy or Copia. In the cases of

insufficient domain information, the elements were assigned as still undetermined.

Most of the transposons insert at random locations leading to novel and usually unique sequence stretches at both borders around the inserted element and the neighbouring original sequence. The de novo detected full-length LTR set provides defined element borders, a prerequisite for the exact positioning of transposable element junctions. We used 100-bp single transposable element junctions with 50 bp outside and 50 bp inside the element from both sides of the element and merged them to 200 bp joined junctions per element. Junctions from the reverse strand were reverse-complemented. The 200-bp joined junctions from all 20 lines were clustered with vmatch dbcluster (http://www.vmatch.de, version 2.3.0) at 98% identity and 98% mutual length coverage (command-line parameters: 98 98 -e 2 -l 98 -d). About 97% of the clusters belonged to the 1:1 type with a maximum of 1 member per line and were used for the downstream analyses. Using the above-described 200-bp joined junctions instead of full sequences reduces the amount of data for clustering to 2%, from about 10 kb to 200 bp per full-length LTR element, thus allowing a sequence clustering of 2.2 million elements in the first place. By including sequence information outside of the element, the repetitiveness of high-copy transposable element families is removed and at the same time the syntenic context is provided even for elements located on chrUn (that is, not assigned to chromosomal pseudomolecules).

### PAV detection and validation
Owing to higher sensitivity in detecting deletions over insertions, a paired genome alignment strategy was used in which each assembly was aligned to reference genome Morex reciprocally by treating Morex as a query and reference using Minimap2 (v.2.17)[50]. From these two alignments, insertion and deletions were called using Assemblytics (v.1.2.1)[26]. Then, only deletions were selected in both alignments and converted into PAVs with regard to Morex. In addition, a hard filter was used to discard PAVs containing more than 5% gaps (Ns) and nested PAVs. We used a previously described method[51] to map deletions longer than 5 kb in Barke relative to Morex using whole-genome shotgun data for 90 Morex × Barke recombinant inbred lines[19]. Mosdepth (v.0.2.9)[52] was used for determining read depth in genomic intervals.

### k-mer-based genome-wide association
PAVs overlapping with single copy regions were identified by BedTools (v.2.28.0)[53]. k-mer sequences with step size of 2 bp were retrieved from single-copy regions residing within PAVs. The abundances of the extracted k-mer sequences were counted in sequence reads using BBDuk (BBMap_37.93) (https://sourceforge.net/projects/bbmap/). k-mer counts were obtained for whole-genome shotgun data of 300 diverse varieties of barley generated in the present study and previously published genotyping-by-sequencing data[9]. k-mer counts were imported into R (v.3.5.1)[54] and normalized for differences in read depth between samples. The normalized k-mer counts were then used for genome-wide association scans using GAPIT3[30] and PCA using standard R functions.

### Construction of single-copy pan-genome
To identify single-copy regions in each genome, genomic regions covered by 31-mers occurring more than once were masked using BBDuk (BBMap_37.93)[55]. Based on masking, single-copy regions in each assembly were obtained in .bed format and subsequently related sequences were retrieved using BEDTools (v2.28.0)[53]. Single-copy sequences from all the assemblies were combined to perform an all-against-all blast search. The blast results were filtered (>90% identity and minimum 80% alignment length) and then clustered using the igraph package[56]. A representative from each cluster (the largest contained sequence) was selected and used for estimating pan-genome size. Clusters shared by all the 20 accessions are referred to as the core genome, and clusters

with sequences originating from 1 to 19 genotypes are considered as the variable genome.

### Hi-C library preparation, sequencing and inversion calling
In situ Hi-C libraries were prepared from one-week-old seedlings of barley IPK core50 collection[9] (Supplementary Table 5) based on a previously described protocol[43] Sequencing, Hi-C raw data processing and inversion calling were performed as previously described[34] using the MorexV2 reference genome sequence assembly[6]. The breakpoint regions were identified by pairwise genome alignment using Minimap2 (v.2.17)[50] and PipMaker (http://pipmaker.bx.psu.edu/cgi-bin/pipmaker?basic)[57].

### Resequencing, SNP calling and PCA
Raw reads (Supplementary Table 4) were trimmed with cutadapt (v.1.15) and aligned to the MorexV2 genome assembly[6] using Minimap2 (v.2.17)[50]. The alignments were sorted using Novosort (V3.06.05) (http://www.novocraft.com). BCFtools (v.1.8)[58] was used to call SNPs and short insertions and deletions (indels). The resulting VCF file was converted into Genomic Data Structure (GDS) format using SeqArray package[59] in R to obtain a SNP matrix. Finally, hard filtering was applied to remove SNPs having more than 10% missing data and heterozygosity. Previously generated genotyping-by-sequencing data[9] were aligned to the MorexV2 reference and identified SNPs using a previously described variant calling pipeline[9]. PCAs were performed using snpgdsPCA() function of the package SNPrelate[60].

### RGT Planet × Hindmarsh mapping population
A cross was made between RGT Planet (maternal plant) and Hindmarsh (pollen donor). In total, 38 $F_2$ plants from the direct cross and 233 individual heads from $F_3$ seeds were progressed to the F6 generation by single seed descent method. The $F_6$ recombinant inbred lines (RIL) (224 in total) were used for construction of a genetic linkage map. Genomic DNA was extracted from the leaves of a single plant per RIL using the cetyl-trimethyl-ammonium bromide method. DNA quality was assessed on 1% agarose gels and quantified using a NanoDrop spectrophotometer (Thermo Scientific NanoDrop Products). DNA was diluted into 50 ng/µl and placed in a 96-well plate for PCR. DArT-seq genotyping-by-sequencing was performed using the DArT-seq platform (DArT PL) according to the manufacturer's protocol (https://www.diversityarrays.com/). In brief, 100 µl of 50 ng µl$^{-1}$ DNA was sent to DArT PL, and genotyping-by-sequencing was performed using complexity reduction followed by sequencing on a HiSeq Illumina platform as previously described[61] (Supplementary Table 9). Sequences flanking polymorphisms detected by DArT-seq were aligned against the MorexV2 genome assembly to determine their physical positions (Supplementary Table 7).

### Field experiments and phenotypic data
Field experiments were conducted at six sites: Gibson, Western Australia (WA, −33.612176, 121.798438); Williams, Western Australia (−33.577668, 116.734934); Wongan Hills, Western Australia (−30.848953, 116.756461); Merredin, Western Australia (−31.487009, 118.229668); South Perth, Western Australia (−31.991186, 115.887944); and Shepperton, Victoria (−36.487551, 145.388470). The distance between South Perth and Shepperton is over 3,300 km. The Merredin site is located inland and receives little rainfall, whereas the Gibson site receives a high amount of rainfall: the other sites are in between. The experimental design for field trial sites was performed as previously described[62]. In brief, all regional field trials (partially replicated design) were planted in a randomized complete block design using plots of 1 by 5 m$^2$, laid out in a row−column format and the middle 3 m was harvested for grain yield. Field trials in South Perth and Shepperton were conducted using double rows with a 40-cm distance within and between rows, owing to space constraints. Seven control varieties were used for spatial adjustment of the experimental

data. Measurements were taken at each plot of each field experiment in the study to determine flowering time (days to Zadoks stage (ZS)49), plant height and grain yield. In brief, heading date was recorded as the number of days from sowing to 50% awn emergence above the flag leaf (ZS49), as a proxy for flowering time. Plant height was determined by estimating the average height from the base to the tip of the head of all plants in each plot. Grain yield (kg ha⁻¹) was determined by destructively harvesting all plant material from each plot to separate the grain, and then determining grain mass. Grain yield data of the field experiments, as well as plant height and heading data, were analysed using linear mixed models in ASReml-R (https://www.vsni.co.uk/software/asreml-r/) to determine best linear unbiased predictions or best linear unbiased estimations for each trait for further analysis. Local best practices for fertilization and disease control were adopted for each trial site.

## Quantitative trait loci (QTL) mapping
Software MapQTL6 was used for the QTL analysis[63]. The genotypic data, phenotypic data and genetic map were formatted and imported to MapQTL6. Interval mapping was conducted for each trait, and then the markers with a logarithm of odds (LOD) value of above 3.0 were selected as cofactors. Multiple QTL model mapping was performed to re-calculate the QTL. If the markers with the highest LOD value were inconsistent with the cofactor markers, then the new markers were selected as cofactors and re-calculated. The QTL results and charts were exported from the software.

## Long-read sequence assembly of the Morex cultivar
PacBio libraries were constructed using SMRTbell Template Prep Kit 1.0 and sized on a SAGE Blue Pippin instrument 20–80 kb. Sequencing was performed on Sequell II device at the HudsonAlpha Institute using V.1.0 chemistry and 10-h movie time. Data were generated from a total of five SMRT cells, yielding 604 Gb of raw sequence reads. A total of 520.72 Gb of this set (104.15×) was used for assembly (Supplementary Tables 11, 12). Previously published Illumina short-read data (ERR3183748 and ERR3183749[6] (Supplementary Table 12)) was used for polishing and error correction. Before use, Illumina fragment reads were screened for phix contamination. Reads composed of >95% simple sequences were removed. Illumina reads shorter than 50 bp after trimming for adaptor and quality ($q < 20$) were removed. The final read set consists of 605,178,701 reads, representing a total of 43.17× of high-quality Illumina bases. The initial assembly was generated by assembling 32,743,478 PacBio reads (104.15× sequence coverage) using MECAT (v.1.1)[64] and subsequently polished using Arrow[65]. This produced an initial assembly of 1,577 scaffolds (1,577 contigs), with a contig N50 of 10.4 Mb, 987 scaffolds larger than 100 kb and a total genome size of 4,139.8 Mb (Supplementary Table 13).

A first round of breaking chimeric scaffolds was done using the POP-SEQ genetic map[19] to identify contigs bearing markers from distant genomic regions. A total of 17 misjoins were identified and resolved. Homozygous SNPs and indels were corrected in the release consensus sequence using a subset of about 30× of the Illumina reads described above in this section. Reads were aligned using BWA-MEM[66]. Homozygous SNPs and indels were discovered with GATK's UnifiedGenotyper tool[67]. A total of 59 homozygous SNPs and 15,759 homozygous indels were corrected. After these correction steps, the assembly contains 4,139.7 Mb of sequence, consisting of 1,594 contigs with a contig N50 of 10.2 Mb. A second round of chimaera breaking by inspecting Hi-C contact matrices as described in the TRITEX pipeline[6]. Published Hi-C data of the Morex cultivar was used[5]. Corrected contigs were arranged into pseudomolecules using TRITEX.

## Comparison of PacBio continuous long read (CLR) and TRITEX assemblies of the Morex cultivar
Full-length cDNA sequences[44] were aligned to the assemblies to assess gene space completeness. Only alignments with query coverage ≥90%

and identity ≥90% were considered. Whole-genome assemblies were done with Minimap2. Structural variant calling with Assemblytics (v.1.2.1)[26] (Morex TRITEX versus Morex CLR; Morex CLR versus Barke) and extraction of single-copy regions were done as described in 'PAV detection and validation'.

## Reporting summary
Further information on research design is available in the Nature Research Reporting Summary linked to this paper.

## Data availability
All raw sequence data collected in this study and sequence assemblies have been deposited at the European Nucleotide Archive (ENA). Accession codes for raw data and assemblies are listed in Supplementary Tables: Supplementary Table 14 (assemblies), Supplementary Table 10 (assembly raw data), Supplementary Table 4 (whole-genome shotgun sequencing), Supplementary Table 5 (Hi-C) and Supplementary Table 9 (DArT-seq). Assemblies, annotations and analysis results were deposited under a DOI in the PGP repository[68] using the e!DAL submission system[69] and are accessible under the URL https://doi.org/10.5447/ipk/2020/24. Assemblies and gene annotations can also be downloaded from https://barley-pangenome.ipk-gatersleben.de. The Barley Pedigree Catalogue is available at http://genbank.vurv.cz/barley/pedigree/.

## Code availability
Source code is released in a public Bitbucket repository, at https://bitbucket.org/ipk_dg_public/barley_pangenome/.

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

**Acknowledgements** We thank M. Knauft, I. Walde and S. König for technical assistance; D. Schüler for sequence data management; J. Bauernfeind, T. Münch and H. Miehe for IT administration; D. Arend for help with data submission; M. Bayer for advice on transcriptome analysis; and M. Herz for pedigree information. This research was supported by grants from the German—Federal Ministry of Education and Research to N.S., M.M., U.S., M.S. and K.F.X.M. (SHAPE, FKZ 031B0190), to U.S. and K.F.X.M. (de.NBI, FKZ 031A536) and to N.S. (COBRA, FKZ 031A323A); the Australian Grain Research and Development Cooperation (9176507) to C.L., K.C., P.L. and P.W.; JST CREST Japan (no. JPMJCR16O4 to K.M. and T.H.); JST Mirai Program Japan (no. 18076896 to K.S.); the National Key R&D Program of China (2018YFD1000701 and 2018YFD1000700) to D.X. and J.Z.; by funding from the China Agriculture Research System (CARS-05) and the Agricultural Science and Technology Innovation Program to C.W. and G.G. Support for 10X sequencing was provided by a research grant from Genome Canada and Genome Prairie to C.P. and J.E.; and by the Natural Science Foundation of China (31620103912) and the National Key R&D Program of China (2018YFD1000706) to G.Z. We acknowledge support from the European Research Council (ERC Shuffle, project identifier: 66918) to R.W.

**Author contributions** N.S. and M.M. designed the study. N.S. coordinated experiments and sequencing. M.M. supervised sequence assembly. M. Spannagl and K.F.X.M. supervised annotation. U.S. supervised data management and submission. S.P., A.H., J.E., D.X., L.B.B. and J.G. performed sequencing experiments. M.J., C.M., Y.G., C.P., J.J. and J.S. performed sequence assembly. M.J. performed structural variation and genome-wide association scan analysis. A.F. submitted sequence data. G.H., T.L., H.G., V.S.B., N.K. and D.L. annotated and analysed genes and transposable elements. S.P., M.J., X.-Q.Z., T.T.A., G. Zhou, C.T., C.H., P.W., M.M. and C.L. analysed polymorphic inversions. H.B., J.G., J.S., J.Z., C.W., G.G., G. Zhang, K.M., T.H., K.S., K.J.C., P.L., C.J.P., C.L., M. Schreiber, R.W. and N.S. contributed sequence data. M.J., S.P., C.L. and M.M. wrote the paper with input from all co-authors.

**Competing interests** The authors declare no competing interests.

**Additional information**
**Correspondence and requests for materials** should be addressed to C.L., M.M. or N.S.

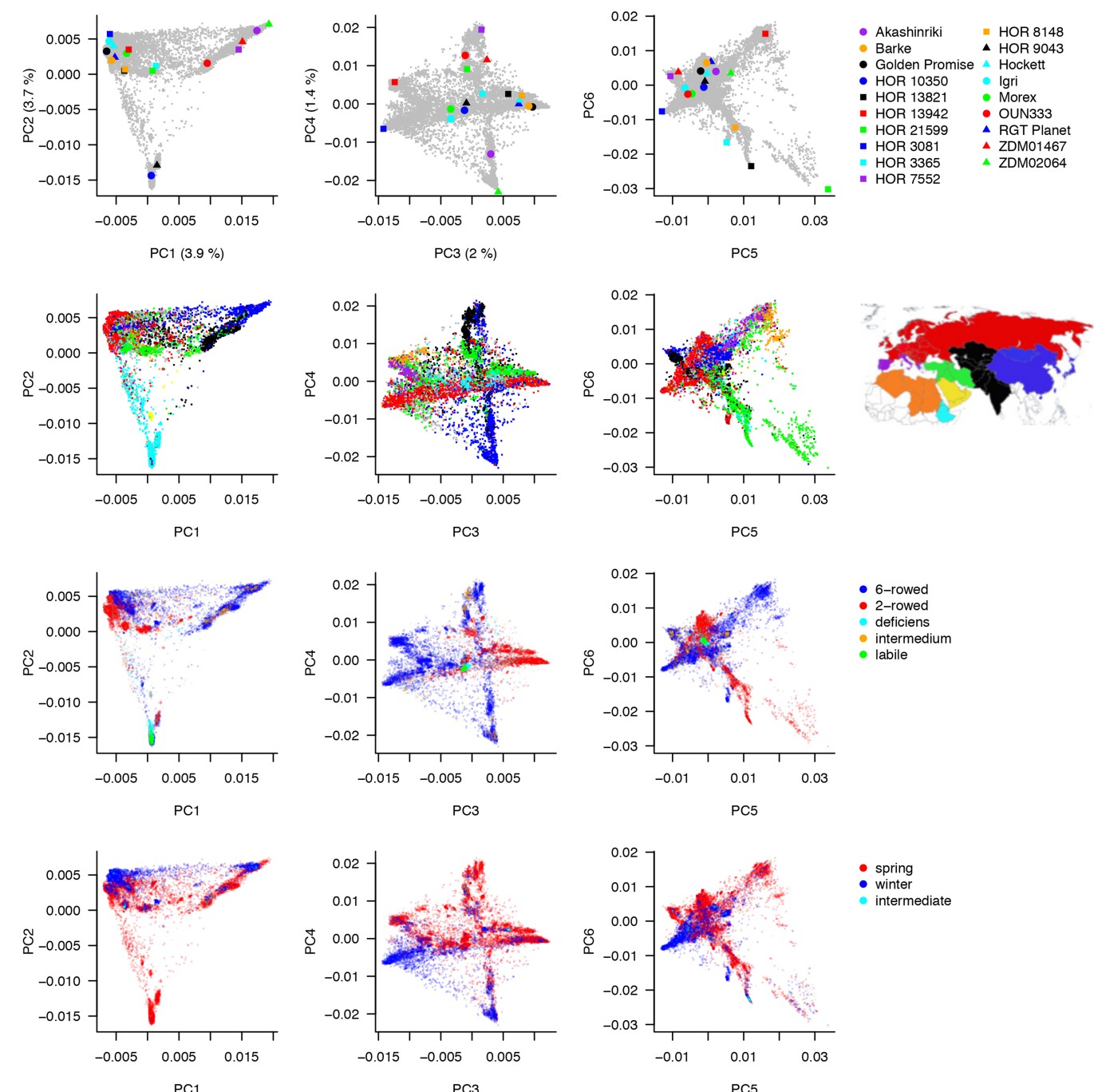

**Extended Data Fig. 1 | Pan-genome selection in the global barley diversity space.** PCA with genotyping-by-sequencing data of 19,778 varieties of domesticated barley sampled from the gene bank of the IPK[9]. The first six principal components are shown. Samples are coloured to highlight the pan-genome selection (first row), or according to geographic origin (second row), row type (third row) or annual growth habit (fourth row). The proportion of variance explained by the principal components is indicated in the axis labels of the first row. The map was created with the R package mapdata[54].

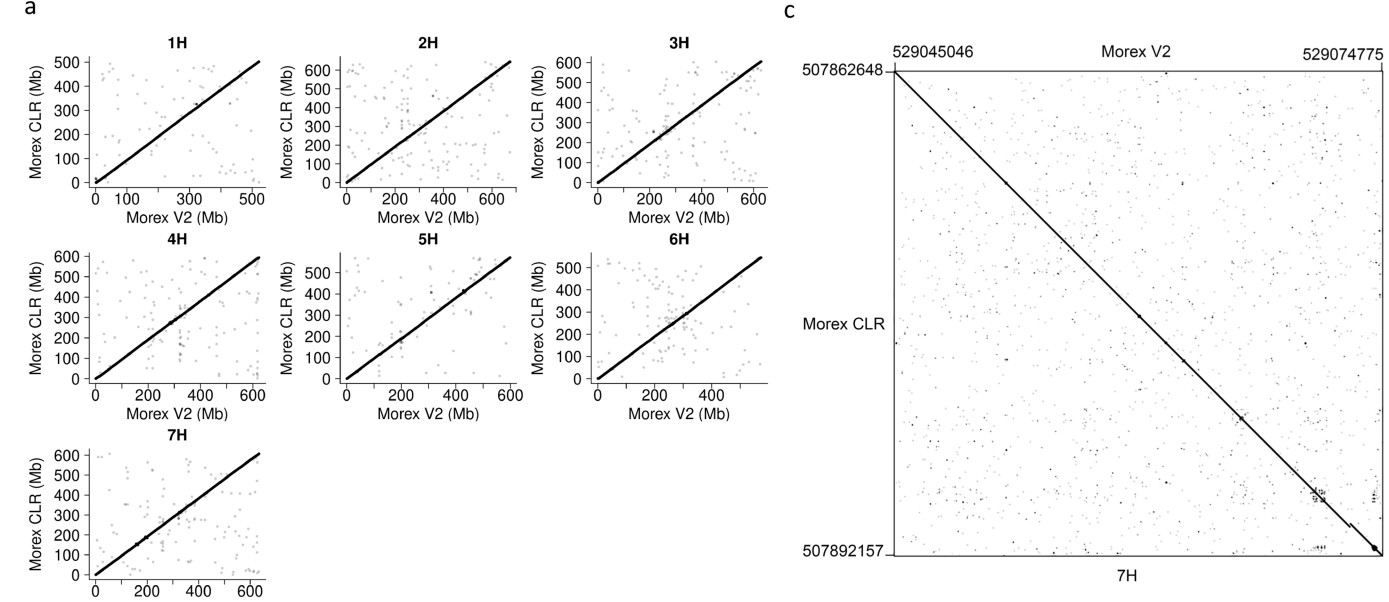

**a**

**b**

| | Morex V2 | Morex CLR |
|---|---|---|
| Pseudomolecule size# | 4,257,712,555 | 4,072,877,080 |
| contig N50 | 32.7 kb | 10.2 Mb |
| contig N90 | 1.4 kb | 1.9 Mb |
| Bionano label sites (%) | 90.5 | 93.1 |
| % cDNA alignment | 91.0 | 89.5 |

# 1H-7H

**d**

| | No. of PAV (> 1kb) | Total size | Mean PAV size |
|---|---|---|---|
| Insertion | 648 | 7.3 Mb | 11.3 kb |
| Deletion | 6,162 | 10 Mb | 1.6 kb |
| Repeat expansion | 4,610 | 152 Mb | 32.9 kb |
| Repeat contraction | 5,102 | 95 Mb | 18.5 kb |
| Tandem expansion | 0 | 0 | 0 |
| Tandem contraction | 1,643 | 6.4 Mb | 3.8 kb |

Reference: Morex V2; Query: Morex CLR

**e**

| Reference | Query | No. PAVs (> 1 kb) | Presence# | Absence# | Single-copy overlapping presence | Single-copy overlapping absence |
|---|---|---|---|---|---|---|
| Morex V2 | Barke | 14,636 (66.8 Mb) | 7,013 (38.5 Mb) | 7,623 (28.3 Mb) | 3,378 (5.4 Mb) | 4,686 (5.5 Mb) |
| Morex CLR | Barke | 15,532 (87.4 Mb) | 8,226 (44.2 Mb) | 7,306 (43.2 Mb) | 3,598 (5.6 Mb) | 4,232 (5.4 Mb) |

#Presence: Present in query genome; #Absence: Absent in query genome

**Extended Data Fig. 2 | Comparison between long-read and short-read assemblies of the Morex cultivar. a**, Co-linearity between Morex V2 (short-read) assembly and the Morex PacBio CLR assembly at the pseudomolecule level. **b**, Summary statistics of the Morex PacBio CLR assembly and Morex V2 assembly. **c**, Alignment of *NUDUM* locus (16 kb) between Morex PacBio CLR and Morex V2. **d**, Structural variants between Morex V2 and Morex PacBio CLR assemblies as detected and classified by Assemblytics. **e**, PAVs between Barke and the Morex V2 and Morex CLR assemblies.

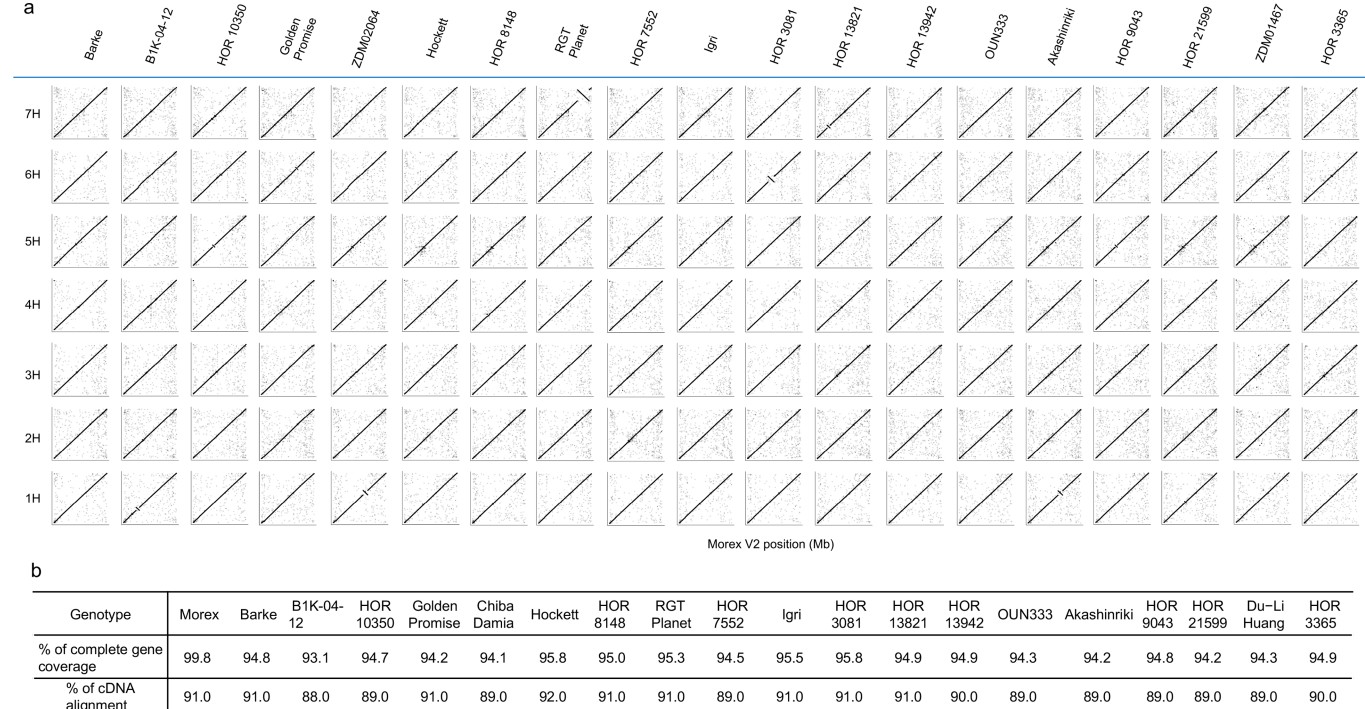

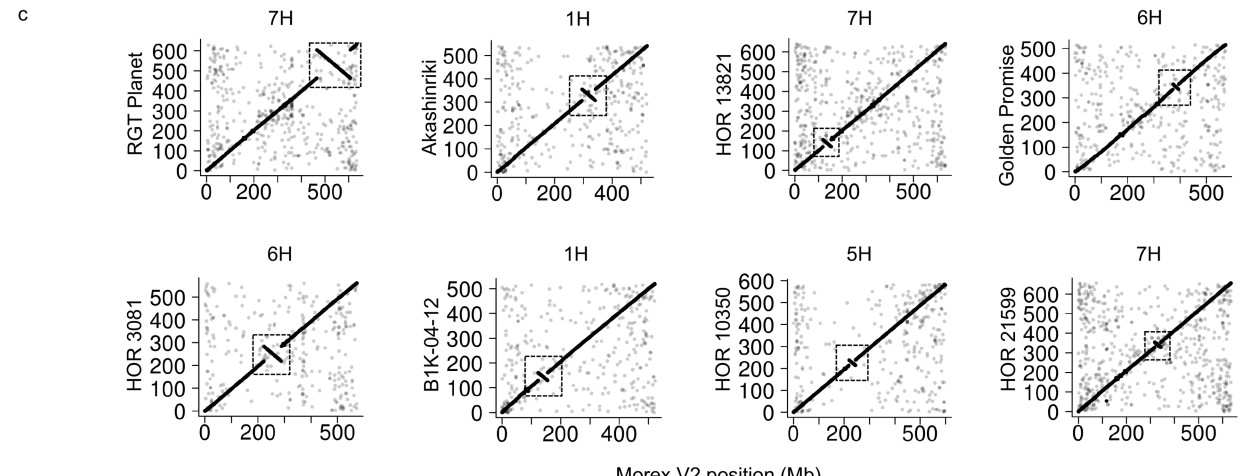

| Genotype | Morex | Barke | B1K-04-12 | HOR 10350 | Golden Promise | Chiba Damia | Hockett | HOR 8148 | RGT Planet | HOR 7552 | Igri | HOR 3081 | HOR 13821 | HOR 13942 | OUN333 | Akashinriki | HOR 9043 | HOR 21599 | Du-Li Huang | HOR 3365 |
|---|---|---|---|---|---|---|---|---|---|---|---|---|---|---|---|---|---|---|---|---|
| % of complete gene coverage | 99.8 | 94.8 | 93.1 | 94.7 | 94.2 | 94.1 | 95.8 | 95.0 | 95.3 | 94.5 | 95.5 | 95.8 | 94.9 | 94.9 | 94.3 | 94.2 | 94.8 | 94.2 | 94.3 | 94.9 |
| % of cDNA alignment | 91.0 | 91.0 | 88.0 | 89.0 | 91.0 | 89.0 | 92.0 | 91.0 | 91.0 | 89.0 | 91.0 | 91.0 | 91.0 | 90.0 | 89.0 | 89.0 | 89.0 | 89.0 | 89.0 | 90.0 |

**Extended Data Fig. 3 | Assessment of contiguity and completeness in 20 genome assemblies. a**, Whole-genome alignments of assemblies of 19 diverse barley accessions to the Morex V2 reference assembly. **b**, Alignment summary of full-length coding sequences (32,878) from the MorexV2 annotation and full-length cDNAs (28,622 full-length cDNAs) in each assembly. Alignments with less than 90% query coverage and 97% (less than 90% for full-length cDNAs) identity were discarded. **c**, Whole-genome alignments show some examples of large chromosomal inversions identified using Hi-C data.

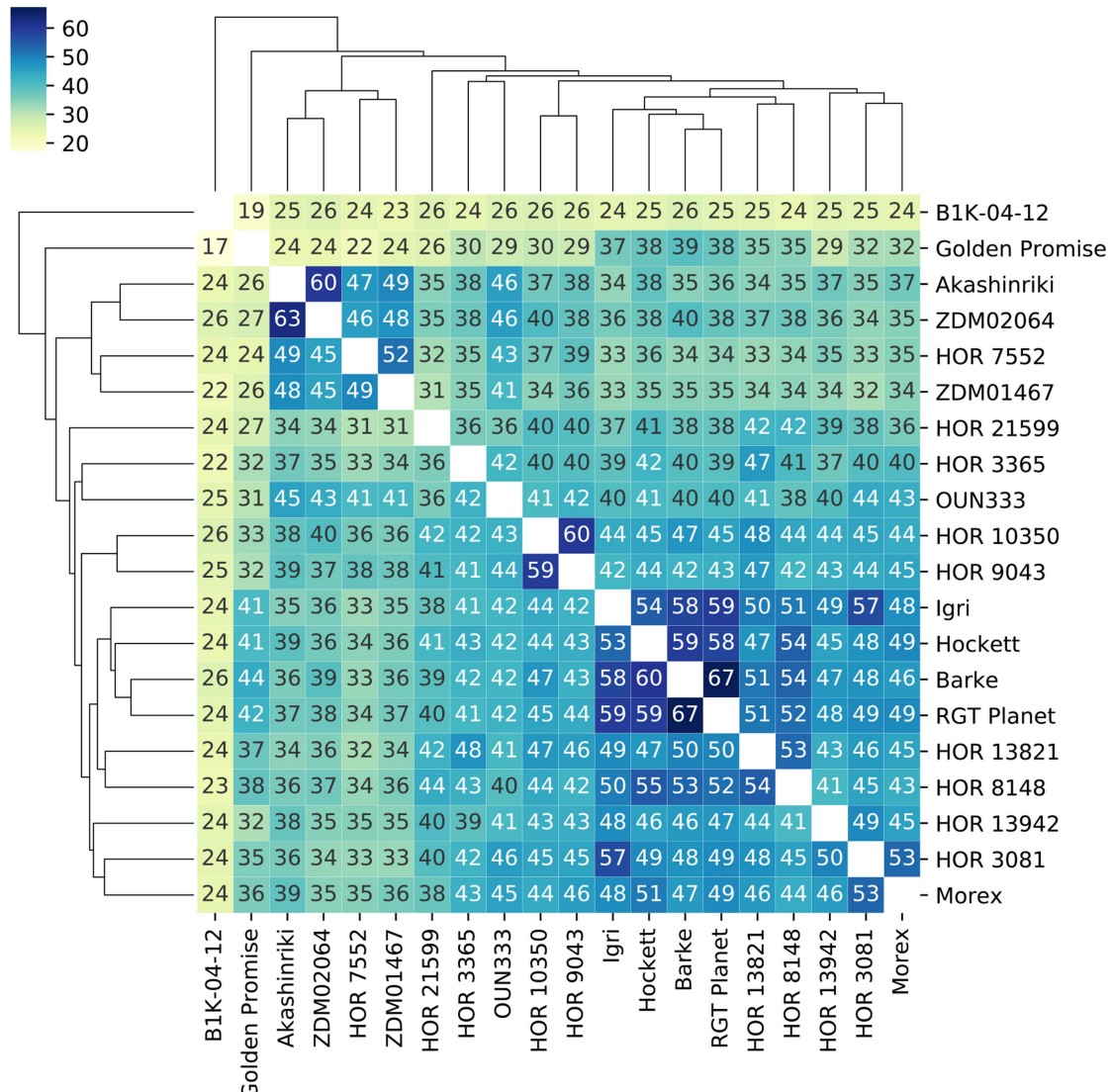

**Extended Data Fig. 4 | Pairwise shared syntenic full-length LTR locations.** The wild variety B1K-04-12 is set apart as an outgroup, as it shares only 19–26% of its still-intact full-length LTR positions with the other landraces and cultivars. The highest similarity is found between the Barke and RGT Planet cultivars (67% shared full-length LTRs).

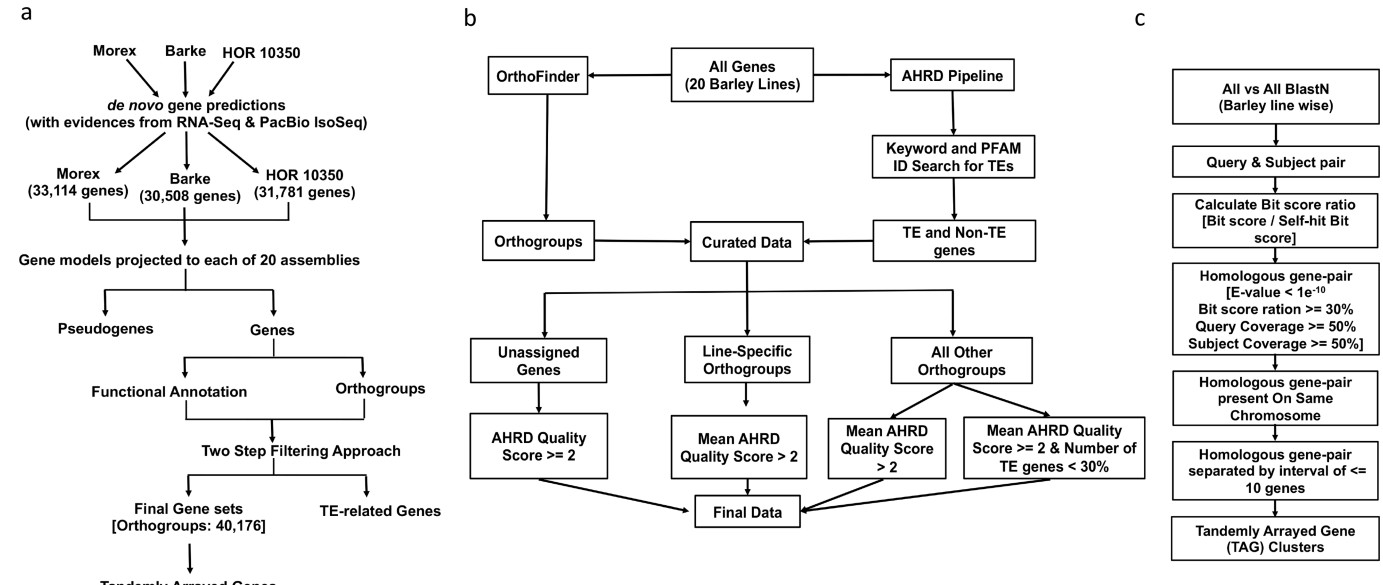

d

| Accession | No. of projected genes | Annotated by AHRD | No. of genes in OGs | No. of non-TE Genes | No. of TAG Clusters | No. of genes in TAG cluster |
|---|---|---|---|---|---|---|
| Akashinriki | 44446 | 44431 | 44278 | 36948 | 2050 | 5262 |
| Barke | 45999 | 45990 | 45858 | 38302 | 2260 | 6115 |
| Chiba Damia | 45050 | 45037 | 44747 | 37292 | 2154 | 5460 |
| B1K-04-12 | 44566 | 44550 | 43179 | 36366 | 2036 | 5420 |
| Golden Promise | 42464 | 42452 | 42159 | 35859 | 1891 | 4700 |
| HOR 3365 | 47588 | 47571 | 47359 | 40044 | 2505 | 6397 |
| Hockett | 46450 | 46434 | 46166 | 38725 | 2398 | 5998 |
| HOR 10350 | 45810 | 45800 | 45541 | 38074 | 2227 | 5981 |
| HOR 13821 | 44714 | 44701 | 44546 | 37199 | 2064 | 5269 |
| HOR 13942 | 44718 | 44706 | 44502 | 37252 | 2039 | 5162 |
| HOR 21599 | 44456 | 44442 | 44079 | 36834 | 2007 | 5138 |
| HOR 3081 | 45146 | 45135 | 44967 | 37502 | 2095 | 5494 |
| HOR 7552 | 44641 | 44622 | 44349 | 37047 | 2080 | 5389 |
| HOR 8148 | 45026 | 45012 | 44894 | 37474 | 2106 | 5522 |
| HOR 9043 | 45028 | 45017 | 44889 | 37459 | 2103 | 5357 |
| Du Li Huang | 44746 | 44727 | 44532 | 37116 | 2119 | 5277 |
| Igri | 45213 | 45202 | 45065 | 37590 | 2101 | 5544 |
| Morex | 46294 | 46282 | 45939 | 38352 | 2170 | 5756 |
| OUN333 | 44699 | 44682 | 44437 | 37042 | 2058 | 5300 |
| RGT Planet | 45413 | 45399 | 45282 | 37822 | 2165 | 5691 |

e

| Genotype | De novo annotation | Gene projection |
|---|---|---|
| Morex | 33115 | 38352 |
| Barke | 30508 | 38302 |
| HOR 10350 | 31781 | 38074 |

**Extended Data Fig. 5 | Gene projection and transposable element annotation. a**, Schematic of the gene projection workflow. TE, transposable element. **b**, Pipeline for annotation and removing transposable elements. **c**, Steps to identify tandemly arrayed gene (TAG) clusters in each assembly. **d**, Summary of gene projections and transposable element annotation in 20 accessions. **e**, Comparison between de novo annotations and gene projections for three genotypes. Reported counts refer to non-transposable-element genes.

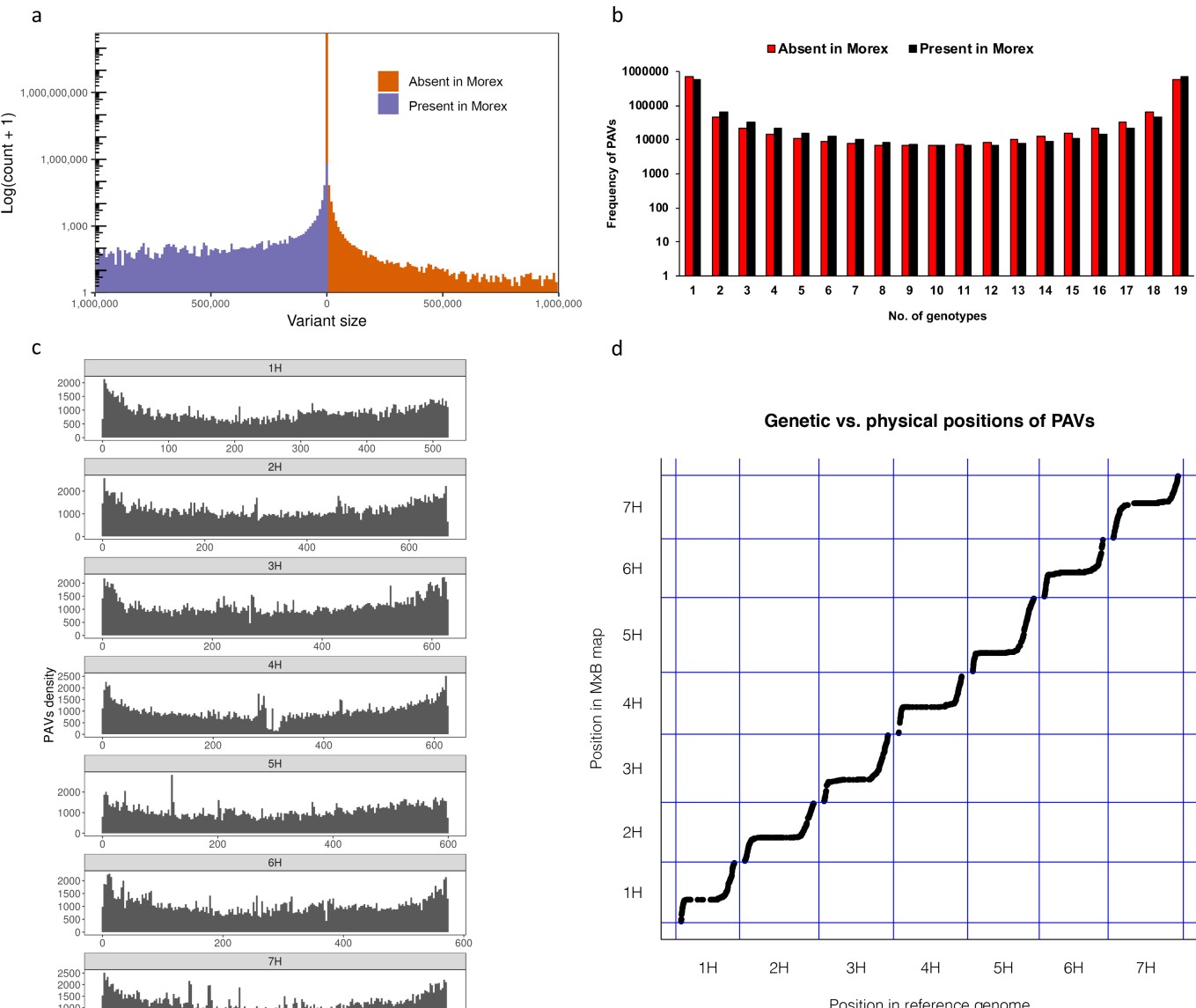

**Extended Data Fig. 6 | Summary of PAVs detected in pan-genome assemblies. a**, Size distribution of PAVs. **b**, Number of PAVs between 20 genome assemblies. **c**, Distribution of PAVs along the barley genome.

**d**, Co-linearity between physical position of PAVs detected between the Morex and Barke cultivars, and mapped genetically in the POPSEQ population.

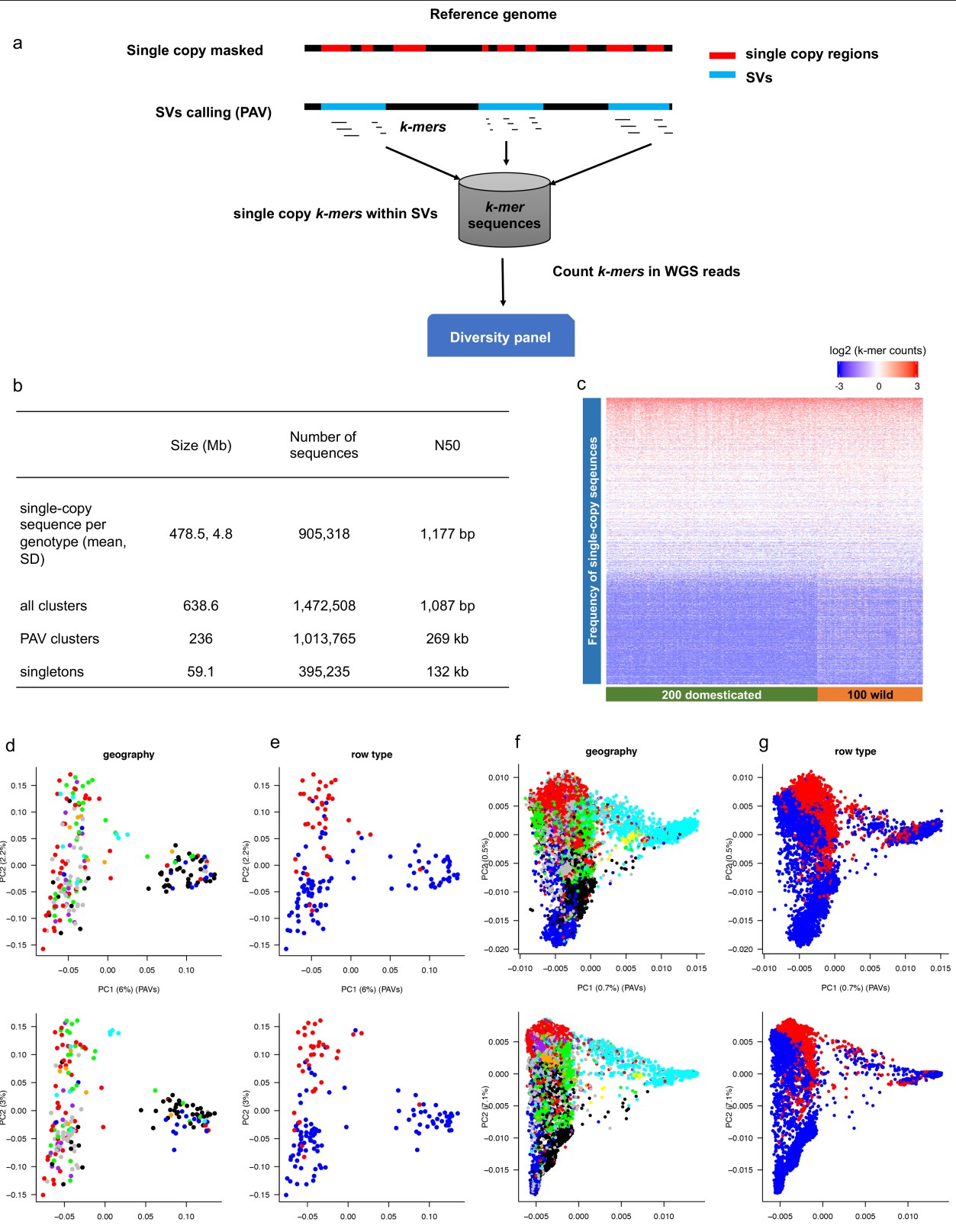

**Extended Data Fig. 7 | Analysis of the single-copy pan-genome. a**, Pipeline used to select single-copy *k*-mers in PAVs as markers for genome-wide association scan analysis. **b**, Summary of single-copy sequence in 20 genome assemblies and results of their clustering. **c**, Copy number of single-copy sequences in a diversity panel comprising 200 domesticated and 100 wild accessions. Frequency ranges from blue (low) to red (high). **d**–**g**, Comparison of PCA on the basis of PAV and SNP variants in whole-genome shotgun data of 200 diverse accessions (**d**, **e**) and 19,778 varieties of domesticated barley[9] (**f**, **g**). Top panels show PCA results from 160,716 PAVs; bottom panels show PCA results from 779,503 of genotyping-by-sequencing SNPs. The accessions are coloured according to geographical origin and row type (using the colour code defined in Extended Data Fig. 1).

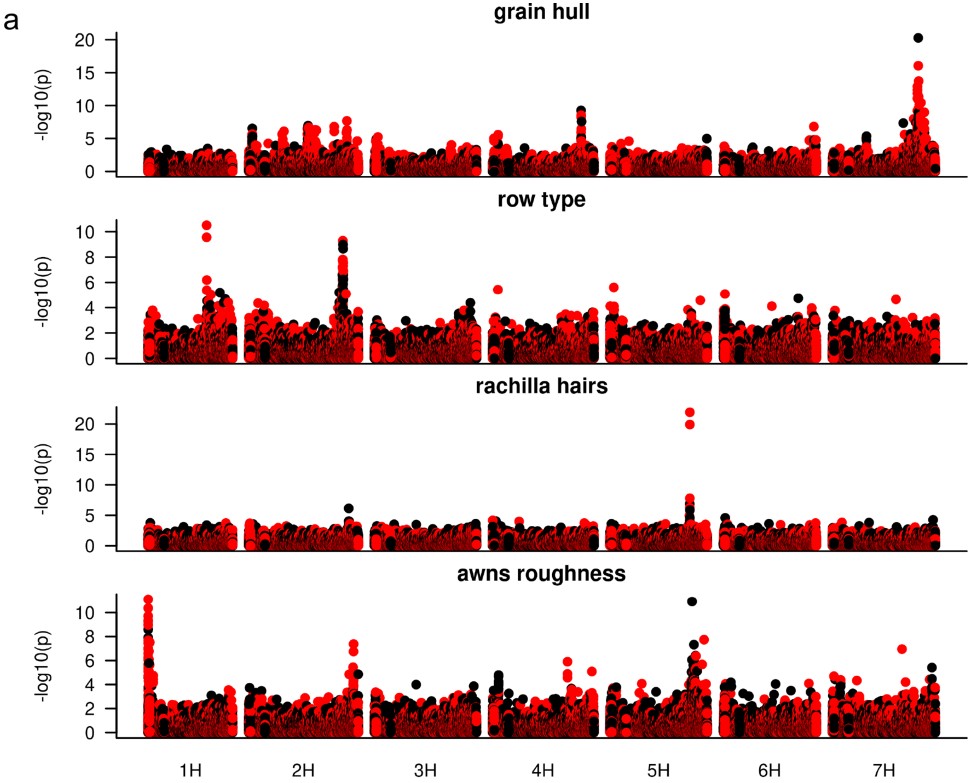

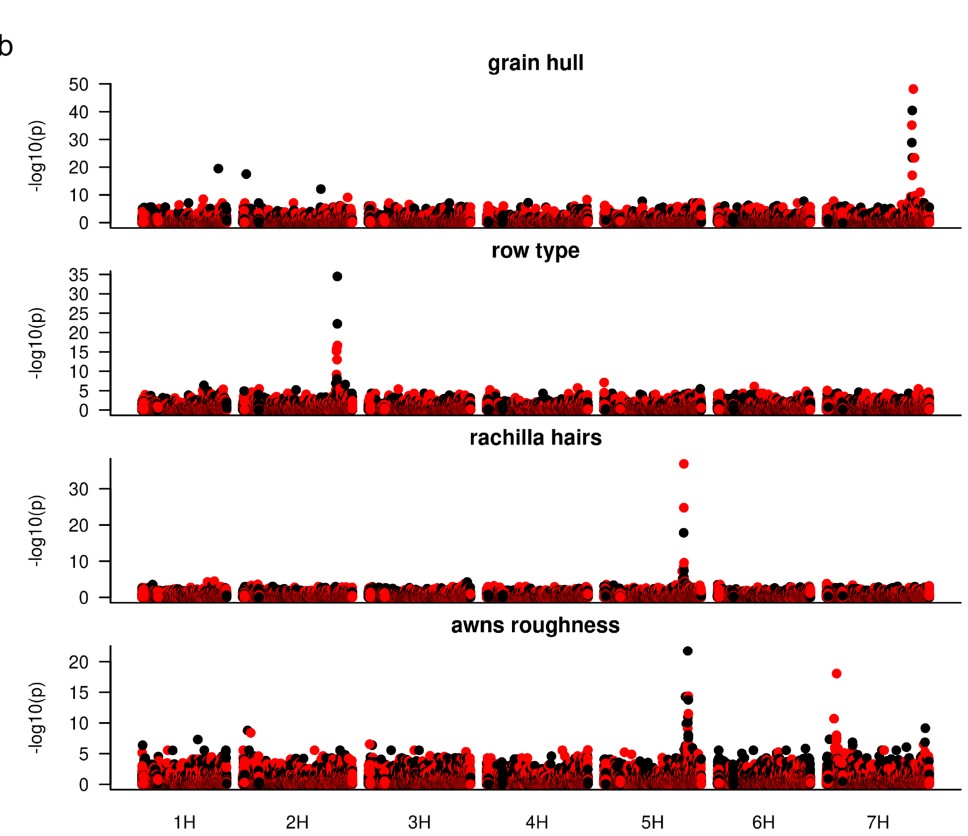

**Extended Data Fig. 8 | PAV-based genome-wide association scans using whole-genome shotgun and genotyping-by-sequencing data. a**, Manhattan plots of PAV-based genome-wide association scans for morphological traits, including adherence of grain hull, row type, length of rachilla hairs and awn roughness, using whole-genome shotgun data from 200 diverse varieties of domesticated barley. **b**, PAV-based genome-wide association scan results for these traits using genotyping-by-sequencing data from 1,000 diverse varieties of domesticated barley collected from the gene bank of the IPK[9]. The 200 varieties of barley used for whole-genome shotgun sequencing are a subset of the 1,000 genotyping-by-sequencing genotypes.

a

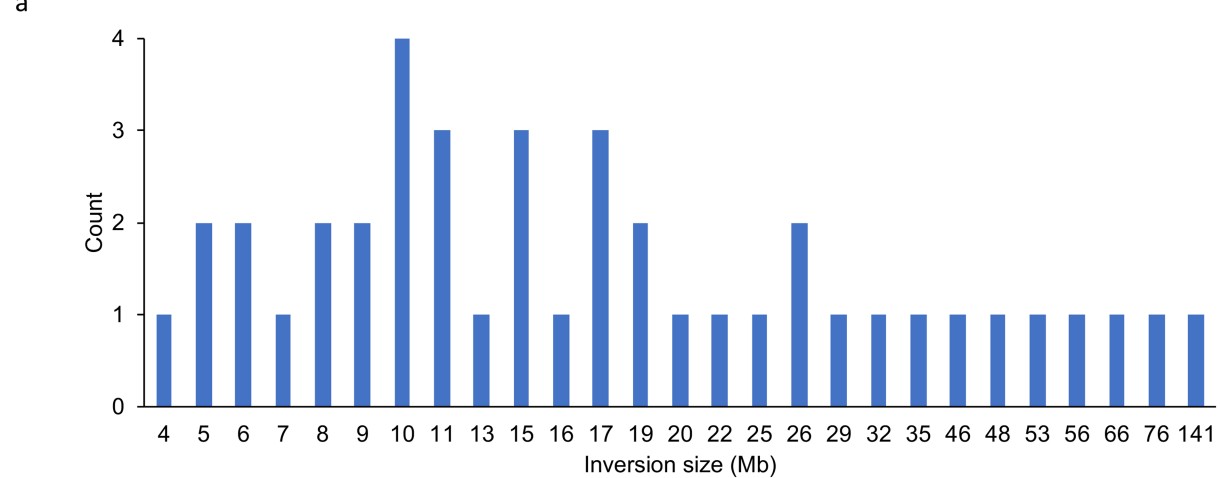

b

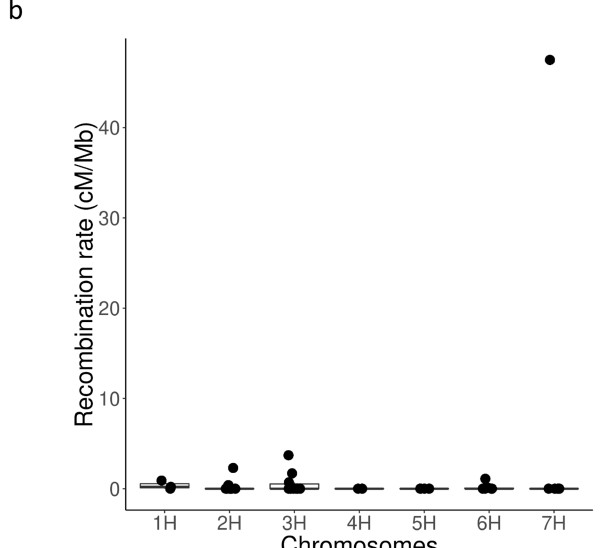

c

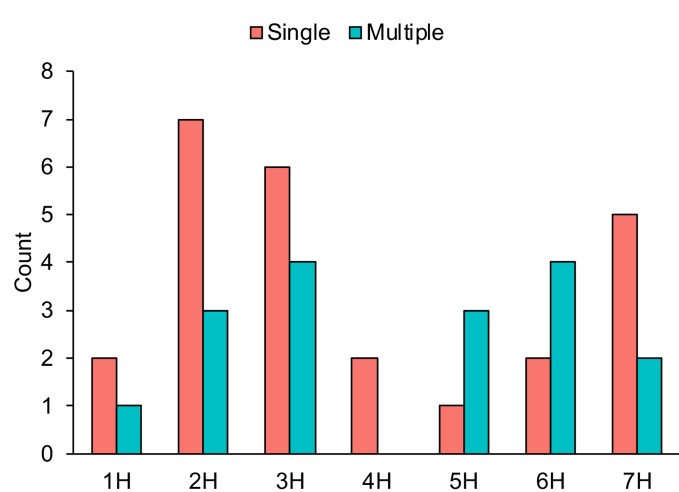

**Extended Data Fig. 9 | Characterization of large inversions in barley.**
**a**, Inversion size distribution. **b**, Recombination in inverted regions.
Recombination rate was determined in the Morex × Barke RIL population[19]

($n$ = 90 genotypes). **c**, Number of inversions present as singletons or shared
between two or more accessions on each chromosome.

**Extended Data Table 1 | Summary statistics of 20 pan-genome assemblies and annotation**

| Accession | Status | Row-type | Growth habit | Country of orgin | No. of super-scaffolds# | Super-scaffolds N50 (Mb) | Size (Gb)# | No. of projected gene models§ | % of transposons | single-copy sequence (Mb)# | Single-copy unique sequence (Mb) |
|---|---|---|---|---|---|---|---|---|---|---|---|
| Akashinriki | cultivar | 6-rowed | winter | Japan | 345 | 34.3 | 4.4 | 36,948 | 80.2 | 475.8 | 1.9 |
| B1K-04-12 | wild | 2-rowed | spring | Israel | 347 | 32.5 | 4.2 | 36,366 | 80.4 | 478.9 | 13.2 |
| Barke | cultivar | 2-rowed | spring | Germany | 284 | 34.8 | 4.1 | 38,302 | 80.3 | 478.3 | 1.5 |
| Golden Promise | cutlivar | 2-rowed | spring | Europe | 1595 | 18.4 | 3.8 | 35,859 | 80.1 | 467.9 | 1.4 |
| Hockett | cultivar | 2-rowed | spring | USA | 482 | 18.4 | 4.0 | 38,725 | 79.9 | 450.0 | 1.6 |
| HOR 10350 | landrace | 6-rowed | spring | Ethiopia | 273 | 30.5 | 4.1 | 38,074 | 80.3 | 477.7 | 2.4 |
| HOR 13821 | landrace | 2-rowed | spring | Turkey | 338 | 33.3 | 4.3 | 37,199 | 80.3 | 477.1 | 2.1 |
| HOR 13942 | landrace | 6-rowed | spring | Southern Europe | 492 | 20.2 | 4.2 | 37,252 | 80.3 | 476.9 | 3.2 |
| HOR 21599 | landrace | 2-rowed | winter | Syria | 318 | 39.4 | 4.3 | 36,834 | 80.3 | 478.1 | 4.7 |
| HOR 3081 | cultivar | 6-rowed | winter | Poland | 552 | 18.9 | 4.2 | 37,502 | 80.2 | 479.4 | 2.2 |
| HOR 3365 | landrace | 6-rowed | winter | Russia | 377 | 31.1 | 4.5 | 40,044 | 79.4 | 476.1 | 2.7 |
| HOR 7552 | landrace | 6-rowed | spring | Pakistan | 558 | 17.7 | 4.2 | 37,047 | 80.2 | 477.9 | 3.7 |
| HOR 8148 | landrace | 2-rowed | spring | Turkey | 519 | 18.8 | 4.2 | 37,474 | 80.3 | 478.5 | 2.0 |
| HOR 9043 | landrace | 6-rowed | spring | Ethiopia | 282 | 42.7 | 4.2 | 37,459 | 80.3 | 478.2 | 2.2 |
| Igri | cultivar | 2-rowed | winter | Germany | 646 | 12.5 | 4.2 | 37,590 | 80.2 | 475.9 | 1.6 |
| Morex | cultivar | 6-rowed | spring | USA | 273 | 40.1 | 4.2 | 38,352 | 80.2 | 479.7 | 2.1 |
| OUN333 | landrace | intermedium | intermediate | Nepal | 337 | 35.1 | 4.4 | 37,042 | 80.3 | 477.2 | 3.7 |
| RGT Planet | cultivar | 2-rowed | spring | Australia | 320 | 37.2 | 4.2 | 37,822 | 80.2 | 479.6 | 1.5 |
| ZDM01467 | landrace | 6-rowed | spring | China | 1397 | 5.0 | 4.4 | 37,116 | 80.3 | 461.3 | 2.5 |
| ZDM02064 | landrace | 6-rowed | spring | China | 720 | 10.9 | 4.0 | 37,292 | 80.2 | 475.5 | 2.2 |

#Chromosomes 1H to 7H.
§Non-transposable element models or transposable-element filtered.

# nature research

| | |
|---|---|

# Reporting Summary

Nature Research wishes to improve the reproducibility of the work that we publish. This form provides structure for consistency and transparency in reporting. For further information on Nature Research policies, see Authors & Referees and the Editorial Policy Checklist .

## Statistics

For all statistical analyses, confirm that the following items are present in the figure legend, table legend, main text, or Methods section.

| n/a | Confirmed | |
|---|---|---|
| ☐ | ✘ | The exact sample size (*n*) for each experimental group/condition, given as a discrete number and unit of measurement |
| ✘ | ☐ | A statement on whether measurements were taken from distinct samples or whether the same sample was measured repeatedly |
| ☐ | ✘ | The statistical test(s) used AND whether they are one- or two-sided<br>*Only common tests should be described solely by name; describe more complex techniques in the Methods section.* |
| ✘ | ☐ | A description of all covariates tested |
| ✘ | ☐ | A description of any assumptions or corrections, such as tests of normality and adjustment for multiple comparisons |
| ✘ | ☐ | A full description of the statistical parameters including central tendency (e.g. means) or other basic estimates (e.g. regression coefficient) AND variation (e.g. standard deviation) or associated estimates of uncertainty (e.g. confidence intervals) |
| ☐ | ✘ | For null hypothesis testing, the test statistic (e.g. $F$, $t$, $r$) with confidence intervals, effect sizes, degrees of freedom and $P$ value noted<br>*Give P values as exact values whenever suitable.* |
| ✘ | ☐ | For Bayesian analysis, information on the choice of priors and Markov chain Monte Carlo settings |
| ✘ | ☐ | For hierarchical and complex designs, identification of the appropriate level for tests and full reporting of outcomes |
| ✘ | ☐ | Estimates of effect sizes (e.g. Cohen's *d*, Pearson's *r*), indicating how they were calculated |

*Our web collection on statistics for biologists contains articles on many of the points above.*

## Software and code

Policy information about availability of computer code

| | |
|---|---|
| Data collection | No software was used for data collection. |
| Data analysis | TRITEX assembly pipeline (commit: 7041ff2), NRGene DeNovoMagic v3.0, Minimap2 (version 2.17), Pipmaker (release 2011-08-12-01), GMAP (release 2018-07-04), SAMtools (v1.8), Novosort (V3.06.05), BCFtools (v1.8), R (v3.5.1), Assemblytics (v1.2.1), Isoseq pipeline (SMRTLink v5.0 Isoseq v1.0), blat (v35x1), exonerate (v2.2.0), AHRD (v3.3.3), OrthoFinder (v2.3.1 ), vmatch (2.3.0), genometools (1.5.9), BBDuk (BBMap_37.93), GAPIT (v3), igraph (v1.1.2), SNPRelate (v1.10.2), MapQTL (v6), MECAT (v1.1), Arrow (v2.3.3) |

For manuscripts utilizing custom algorithms or software that are central to the research but not yet described in published literature, software must be made available to editors/reviewers. We strongly encourage code deposition in a community repository (e.g. GitHub). See the Nature Research guidelines for submitting code & software for further information.

## Data

Policy information about availability of data

All manuscripts must include a data availability statement. This statement should provide the following information, where applicable:

- Accession codes, unique identifiers, or web links for publicly available datasets
- A list of figures that have associated raw data
- A description of any restrictions on data availability

All raw sequence data collected in this study and sequence assemblies have been deposited at the European Nucleotide Archive (ENA). Accession codes for raw data and assemblies are listed in supplementary tables: Supplementary Table 14 (assemblies), Supplementary Table 10 (assembly raw data), Supplementary Table 4 (WGS sequencing), Supplementary Table 5 (Hi-C), Supplementary Table 9 (DArT-seq). Assemblies, annotations and analysis results were deposited under a digital object identifier (DOI) in the PGP repository67 using the e!DAL submission system68 and accessible under the URL http://dx.doi.org/10.5447/ipk/2020/24. Assemblies and gene annotations can also be downloaded from https://barley-pangenome.ipk-gatersleben.de. The Barley Pedigree Catalogue is available at http://genbank.vurv.cz/barley/pedigree/.

# Field-specific reporting

Please select the one below that is the best fit for your research. If you are not sure, read the appropriate sections before making your selection.

[✗] Life sciences          [ ] Behavioural & social sciences          [ ] Ecological, evolutionary & environmental sciences

For a reference copy of the document with all sections, see nature.com/documents/nr-reporting-summary-flat.pdf

# Life sciences study design

All studies must disclose on these points even when the disclosure is negative.

| | |
|---|---|
| Sample size | Accessions for genome assembly were chosen to cover the diveristy (PCA) space of ~20,000 domesticated barley genotypes (Milner et al., 2019 Nature Genetics).<br>Accessions for resequencing (WGS, Hi-C) were chosen based on the core sets defined by Milner et al.<br>No sample size calculation was performed. Samples were chosen from major germplasm groups evident in PCA or because of relevance for barley genetics (Barke, Golden Promise, Igri) |
| Data exclusions | No data were excluded. |
| Replication | A partially replicated design was employed in field trials. No replication was done in genome assembly and sequencing. |
| Randomization | Field trials were done in randomized block designs. Experimental observations were done without predefined groupings. Other forms of randomization were not relevant to this study. |
| Blinding | Blind testing was not done as it was not relevant to plant molecular genetics and genomics work. |

# Reporting for specific materials, systems and methods

We require information from authors about some types of materials, experimental systems and methods used in many studies. Here, indicate whether each material, system or method listed is relevant to your study. If you are not sure if a list item applies to your research, read the appropriate section before selecting a response.

## Materials & experimental systems

| n/a | Involved in the study |
|---|---|
| [✗] | Antibodies |
| [✗] | Eukaryotic cell lines |
| [✗] | Palaeontology |
| [✗] | Animals and other organisms |
| [✗] | Human research participants |
| [✗] | Clinical data |

## Methods

| n/a | Involved in the study |
|---|---|
| [✗] | ChIP-seq |
| [✗] | Flow cytometry |
| [✗] | MRI-based neuroimaging |

