## [Peer Review File · Nature]

Manuscript Title: The barley pan-genome reveals the hidden legacy of mutation breeding

Editorial Notes:**Reviewer Comments & Author Rebuttals****Reviewer Reports on the Initial Version:**Referee #1 (Remarks to the Author):

The authors report a pan genome for barley comprised of 20 individuals and resequencing/genotyping of many, many more accessions. AS far as I am aware, this is the first chromosome-scale pan genome for a crop plant and for a large genome one at that, ~5Gbp. The use of HiC for chromosome-level comparisons allowed them to find and validate structural variants that have not heretofore been possible.

Technical comments: I don't understand why three different sequencing approaches were used (NRGene, TRITEX and W2rap) unless they were provided by different research groups and funded independently. However, the use of HiC for all three assemblies resulted in high quality, chromosome-scale assemblies with similar annotation profiles (TE and gene content). Moreover, the use of orthogonal data sets to confirm the computation analyses (e.g. mapping PAV in More x Barke; PCR genotyping of inversion breakpoints, etc) affirmed to this reviewer that the assemblies were useful for these comparisons/analyses.

The major takeaways for me were 1) the use of Kmers in the single copy genome space to map a gene underlying a phenotype; 2) the instability of genomes as evidenced by naturally occurring TE turnover, small and medium sized inversions and PAV; and 3) the nice example of mutation breeding with a 141 Mbp inversion, likely associated with a phenotype.

I don't have any technical concerns. The paper was well written, the figures logical and the data convincing, especially with the confirmatory analyses backing up the computation predictions.

Referee #2 (Remarks to the Author):

Barley is globally a major crop for food, feed and malting. There are large genetic resources stored in global genebanks and many breeding programs aim to develop varieties that are locally adapted and suited to the specific needs of end users. While the phenotypic diversity of global barley genetic resources is partially known, the molecular diversity is much less characterized. One reason is the absence of high-quality genome sequences that allow to assess molecular diversity such as sequence polymorphisms, structural variation and presence/absence variation (PAV) as well as large-scale inversions. Until now, high quality genome sequences in barley were only available for three reference genomes, which is also due to the large genome size (5 Gbp).

The study of Jayakodi et al. goes beyond these reference genomes and presents the genomes and a comparative analysis of 20 highly diverse barley genotypes. They include elite varieties, landraces and one wild barley accession (the ancestral form of barley from which it was domesticated). The contiguity and sizes of sequence super-scaffolds is the same as for the known reference genomes. Importantly, the 20 sequenced genotypes were carefully selected to represent much of the genetic diversity but also some major phenotypic diversity such as row-type in the

barley gene pool as determined by PCA. The dataset presented in this manuscript represents a significant step toward the pan-genome of barley, i.e. the understanding of the complete set of genes, PAV and structural variation in the gene pool. The authors found a high number of large inversion polymorphisms and characterized in detail two of them. One large inversion was tracked to a likely origin in mutation breeding. Furthermore, the work shows the power of assembling a "single copy pan-genome" as well as PAV for genome-wide association mapping.

This is a landmark study on genome-wide analysis of diversity in a major crop and represents a resource which will transform research in barley genetics as well as molecular breeding for barley improvement. The paper is well written and the data are clearly presented.

A few aspects remain to be clarified in the study:

1. What was the basis for selection of the single wild barley genotype to be included in the dataset? Is it suspected to be close to the genotype(s) involved in domestication? There is also little information in the manuscript on this genotype specifically, and some more comparative analysis of the wild barley genome with the 19 genomes from domesticated barley (in addition to the TE data) would be informative. A short supplementary text might be helpful here.
2. Despite the integration of Hi-C data and 10x Genomics Chromium sequences, some sequence scaffolds may end up in inverted orientation in genome assemblies which can mostly be corrected based on manual curation. Thus, large sequence inversions can be an assembly artefact. The authors should give some indication on the quality control of particularly larger inversions (defined as > 5 Mb by the authors). Furthermore, smaller scale analysis of molecular signatures at the breakpoints of similar rearrangements in wheat have revealed that unequal crossing over and double-strand break repair might be involved in the formation of such diversity. Is there any evidence for molecular mechanisms resulting in the larger inversions in the barley genome?
3. There is a difference of about 10% in gene number/content as annotated by gene projections among the studied barley genotypes (extended data table 1, some information also in Fig.2a). A recent study on the comparative analysis in two wheat genotypes of a single chromosome available in reference genome quality has concluded that gene content across the two chromosomes was highly conserved (Thind et al. Genome Biology 2018). Although a initial comparative analysis by blast suggested a unique gene content between 10-20%, a refined analysis concluded that in fact 99% percent of the genic sequences were conserved. Annotation pipelines might have caused this artefact. The authors should provide some deeper analysis for the conclusions on differences in gene content and reliability of the estimates. It is exactly this problem which makes the single copy pan-genome analysis presented in this paper highly valuable.
4. Line 124: Abundant genic copy number variation between barley genotypes: did the authors study single genetic loci to understand in exemplary cases the diversity of genes and gene organization? It might be useful to have at least one genetic region presented in its diversity between the genotypes.
5. Extended Data Figure 6c: what is the scale, what is the range of frequency from blue to red?

Referee #3 (Remarks to the Author):

This paper on the barley pangenome from 20 "reference" genomes presents some interesting results for genome-enabled breeding efforts. What I am missing, which I think is absolutely critical given the emphasis on structural variation, is verification using long reads of the findings that are here based only on Illumina genomes. These are scaffolded well using 10X and HiC, and seem to be of good (although varied) contiguity, but in this day and age of ready access to PacBio and

Oxford long read technology, I'm afraid I think the bar for the top journal in the world needs to be higher. I'm not at all suggesting the work be redone entirely, but I think it is absolutely necessary that /at least/ one of the accessions be sequenced deep enough to map long reads to its Illumina reference and the pangenome. I'm afraid that all of the points about structural variation leaves the issue open as to how high quality the determinations of SV is. Even highly contiguous 10X/HiC scaffolded assemblies can have extensive collapses in repeat regions and can certainly miss some tandem duplicates. Moreover, it's very hard to assess HiC misjoins without actual molecules to assess contig orientations with. As such, I can't recommend publication until the authors can QC at least one reference using long reads, either ONT or PacBio. This QC should hopefully demonstrate the reliability of the authors' conclusions, and in so doing, make the resources of far greater value to the community. Should QC based on long reads render current results problematic, the authors and editors should reconsider options.

Author Rebuttals to Initial Comments:

Editor

Should further experimental data and analyses allow you to address these criticisms, we would be happy to look at a revised manuscript (unless something similar has been accepted at Nature or appeared elsewhere in the meantime). Specifically, we would strongly encourage you to validate structural variation data, as per reviewer #3's comments.

Answer: To address the comments of reviewer #3, we have performed a detailed comparison between the short-read assembly of barley cv. Morex and a long-read assembly of this genotype, which we have recently constructed from PacBio continuous long reads (CLR). Moreover, we describe the approaches we took for technical validation in this response letter in greater detail than would be possible in the main text.

Referee #1:

The authors report a pan genome for barley comprised of 20 individuals and resequencing/genotyping of many, many more accessions. AS far as I am aware, this is the first chromosome-scale pan genome for a crop plant and for a large genome one at that, ~5Gbp. The use of HiC for chromosome-level comparisons allowed them to find and validate structural variants that have not heretofore been possible.

Technical comments: I don't understand why three different sequencing approaches were used (NRGene, TRITEX and W2rap) unless they were provided by different research groups and funded independently. However, the use of HiC for all three assemblies resulted in high quality, chromosome-scale assemblies with similar annotation profiles (TE and gene content). Moreover, the use of orthogonal data sets to confirm the computation analyses (e.g. mapping PAV in More x Barke; PCR genotyping of inversion breakpoints, etc) affirmed to this reviewer that the assemblies were useful for these comparisons/analyses.

Answer: Indeed, the barley pan-genome is an international research collaboration involving several partners with independent funding. Initially, assemblies were done with different pipelines (NRGene, TRITEX, W2rap) to evaluate them regarding assembly quality, speed and costs. TRITEX was identified

as the most cost-effective method to obtain chromosome-scale assemblies, and was used to assemble most genomes reported in the present study. We note that TRITEX was used as a common pipeline to validate the structural correctness of assembled scaffolds and to arrange them into chromosomal pseudomolecules.

The major takeaways for me were 1) the use of Kmers in the single copy genome space to map a gene underlying a phenotype; 2) the instability of genomes as evidenced by naturally occurring TE turnover, small and medium sized inversions and PAV; and 3) the nice example of mutation breeding with a 141 Mbp inversion, likely associated with a phenotype.

I don't have any technical concerns. The paper was well written, the figures logical and the data convincing, especially with the confirmatory analyses backing up the computation predictions.

Referee #2 (Remarks to the Author):

Barley is globally a major crop for food, feed and malting. There are large genetic resources stored in global genebanks and many breeding programs aim to develop varieties that are locally adapted and suited to the specific needs of end users. While the phenotypic diversity of global barley genetic resources is partially known, the molecular diversity is much less characterized. One reason is the absence of high-quality genome sequences that allow to assess molecular diversity such as sequence polymorphisms, structural variation and presence/absence variation (PAV) as well as large-scale inversions. Until now, high quality genome sequences in barley were only available for three reference genomes, which is also due to the large genome size (5 Gbp).

The study of Jayakodi et al. goes beyond these reference genomes and presents the genomes and a comparative analysis of 20 highly diverse barley genotypes. They include elite varieties, landraces and one wild barley accession (the ancestral form of barley from which it was domesticated). The contiguity and sizes of sequence super-scaffolds is the same as for the known reference genomes. Importantly, the 20 sequenced genotypes were carefully selected to represent much of the genetic diversity but also some major phenotypic diversity such as row-type in the barley gene pool as determined by PCA. The dataset presented in this manuscript represents a significant step toward the pan-genome of barley, i.e. the understanding of the complete set of genes, PAV and structural variation in the gene pool. The authors found a high number of large inversion polymorphisms and characterized in detail two of them. One large inversion was tracked to a likely origin in mutation breeding.

Furthermore, the work shows the power of assembling a "single copy pan-genome" as well as PAV for genome-wide association mapping.

This is a landmark study on genome-wide analysis of diversity in a major crop and represents a resource which will transform research in barley genetics as well as molecular breeding for barley improvement. The paper is well written and the data are clearly presented.

A few aspects remain to be clarified in the study:

1. What was the basis for selection of the single wild barley genotype to be included in the dataset?

Is it suspected to be close to the genotype(s) involved in domestication? There is also little information in the manuscript on this genotype specifically, and some more comparative analysis of the wild barley genome with the 19 genomes from domesticated barley (in addition to the TE data) would be informative. A short supplementary text might be helpful here.

Answer: The genotype B1K-04-12 (alternative name: FT11), a desert ecotype from Ein Prat (Israel), is representative of a highly diverse group of wild barleys from Southern Israel (Hübner et al. 2013 *J Evol Biol*; Russell et al. 2016, *Nat Genet*). We have added this information to the main text (ll. 94-95).

B1K-04-12 is unlikely to have made a special contribution to the domesticated gene pool. The PCA analysis of Russell et al.(2016) rather indicates that it is among the wild barleys that are most distantly related to the domesticate. As a fully interfertile conspecific of domesticated barley, B1K-04-12 is not an outgroup in the phylogenetic sense, but we chose it to get an indication of the extent of structural variation present in wild barley and not found in domesticated barley. As expected, B1K-04-12 had the highest amount of unique sequence in our panel and was diverged from domesticated barley in an analysis of full-length LTRs (Extended Data Figure 4).

2. Despite the integration of Hi-C data and 10x Genomics Chromium sequences, some sequence scaffolds may end up in inverted orientation in genome assemblies which can mostly be corrected based on manual curation. Thus, large sequence inversions can be an assembly artefact. The authors should give some indication on the quality control of particularly larger inversions (defined as > 5 Mb by the authors). Furthermore, smaller scale analysis of molecular signatures at the breakpoints of similar rearrangements in wheat have revealed that unequal crossing over and double-strand break repair might be involved in the formation of such diversity. Is there any evidence for molecular mechanisms resulting in the larger inversions in the barley genome?

Answer: Our pipeline for Hi-C-based pseudomolecule construction (Monat et al. 2019, *Genome Biol*) involves automatic and manual steps (including inspection of Hi-C contact matrices) to detect and correct large (> 5 Mb) misassemblies. Smaller inversions introduced either by 10X or Hi-C scaffolding (1-5 Mb) can be hard to detect from Hi-C contact matrices. The pipeline we used for reference-based inversion detection (Himmelbach et al. 2018, *Plant J*) uses the following approach to avoid false positive inversion calls due to mis-oriented sequences: Hi-C data for the reference genotype (Morex in our case) is used as a control. If a mis-oriented scaffold is present, both the test and the control samples will show deviations in the contact matrix. In this case, no inversion is called. If a true inversion is present, only the test sample will show deviations in the contract matrix. A detailed description of the approaches we took to validate structural variation is given in our response to Reviewer #3.

Schmidt et al. (2019) *Plant J* showed that inversions increase in frequency in mutants deficient in the non-homologous end joining pathway and are induced by micro-homology. The great abundance of highly similar copies of transposable elements in the barley genome make both unequal crossing-over and faulty repair of double strand breaks likely scenarios for the origin of inversions. In the case of the large 7H event we found in cv. RGT Planet, double-strand breaks were likely introduced by irradiation and repaired by ligating distant sequences.

3. There is a difference of about 10% in gene number/content as annotated by gene projections among the studied barley genotypes (extended data table 1, some information also in Fig.2a). A recent study on the comparative analysis in two wheat genotypes of a single chromosome available in reference genome quality has concluded that gene content across the two chromosomes was highly conserved (Thind et al. Genome Biology 2018). Although an initial comparative analysis by blast suggested a unique gene content between 10-20%, a refined analysis concluded that in fact 99% percent of the genic sequences were conserved. Annotation pipelines might have caused this artefact. The authors should provide some deeper analysis for the conclusions on differences in gene content and reliability of the estimates. It is exactly this problem which makes the single copy pan-genome analysis presented in this paper highly valuable.

Answer: We have expanded the description of the gene count differences in the revised manuscript. The number of projected gene models ranges from 35,849 to 40,044 (Extended Data Fig. 5d) with a mean of 37,515 genes, representing an average gene content difference of 2.3 % (standard deviation: 896 gene models, see ll. 123-128 of the revised manuscript). This estimate is lower than the uncorrected estimate of Thind et al. (10-20%), but higher than their conservative 1 % estimate. Our higher number is likely due to the larger genetic diversity captured by our global diversity panel than by the two wheat cultivars in the Thind et al. study (Chinese Spring and CH Campala).

To understand the impact of our projection-based strategy as opposed to de novo annotation, we added a new table (Extended Data Fig. 5e, ll. 125-128 of the revised manuscript), which compares the gene counts in de novo annotations of Morex, Barke and HOR 10350 (based on RNA-seq and Iso-seq for the respective lines) and the corresponding projections. Gene counts in the projections were 16–26 % higher than in the de novo annotations, indicating that some projected genes lack support by transcript data. Possible explanations are low expression due to highly tissue-specific regulation (the Morex RNA-seq data contains more diverse tissues than HOR 10350 and Barke) or pseudogenization. Distinguishing these scenarios would require transcript data for all studied lines. As the reviewer noted, our single-copy pan-genome analyses are not based on gene copy numbers inferred from projections, but on the complete genome sequences. Therefore, over- or underestimates of gene copy number or unclear expression levels of gene models have not influenced the quantitative analysis of pan-genome complexity and associations between SV and agronomic traits. A study obtaining RNA-seq data from multiple tissues of the 20 barley genome accessions are under way as the next stage of the barley pan-genome project, and we refer to this future work in ll. 134-135 of the revised manuscript.

4. Line 124: Abundant genic copy number variation between barley genotypes: did the authors study single genetic loci to understand in exemplary cases the diversity of genes and gene organization? It might be useful to have at least one genetic region presented in its diversity between the genotypes.

Answer: In the revised manuscript, we have expanded the analysis of the *NUDUM* (*NUD*) locus. We used the most highly associated single-copy pan-genome marker (containing the *NUD* gene sequence) and plotted its coverage in our panel of 200 domesticated barleys (Fig. 2d). This analysis

was motivated by a report about another *nud* allele (*nud1.g*) in Tibetan barley. This allele has the complete gene sequence, but protein function is putatively disrupted by a SNP in the coding sequence (Yu et al. 2016, Plant Mol Biol Rep). However, all 36 naked barley in our panel carried the deletion, consistent with the idea that the overwhelming majority of naked barley trace back to a single mutational event. We conclude that wider sampling, including morphologically diverse accessions from centers of (phenotypic) diversity, is required to capture rare variants impacting traits of agronomic relevance (see ll. 186-192 of the revised manuscript).

5. Extended Data Figure 6c: what is the scale, what is the range of frequency from blue to red?

Answer: We have added a color scale to the figure (now Extended Data Fig. 7c). It ranges from -3 and 3 on a log₂ scale.

Referee #3 (Remarks to the Author):

This paper on the barley pangenome from 20 "reference" genomes presents some interesting results for genome-enabled breeding efforts. What I am missing, which I think is absolutely critical given the emphasis on structural variation, is verification using long reads of the findings that are here based only on Illumina genomes. These are scaffolded well using 10X and HiC, and seem to be of good (although varied) contiguity, but in this day and age of ready access to PacBio and Oxford long read technology, I'm afraid I think the bar for the top journal in the world needs to be higher. I'm not at all suggesting the work be redone entirely, but I think it is absolutely necessary that /at least/ one of the accessions be sequenced deep enough to map long reads to its Illumina reference and the pangenome. I'm afraid that all of the points about structural variation leaves the issue open as to how high quality the determinations of SV is. Even highly contiguous 10X/HiC scaffolded assemblies can have extensive collapses in repeat regions and can certainly miss some tandem duplicates. Moreover, it's very hard to assess HiC misjoins without actual molecules to assess contig orientations with. As such, I can't recommend publication until the authors can QC at least one reference using long reads, either ONT or PacBio. This QC should hopefully demonstrate the reliability of the authors' conclusions, and in so doing, make the resources of far greater value to the community. Should QC based on long reads render current results problematic, the authors and editors should reconsider options.

Answer:

We address the reviewer's concerns about the impact of assembly quality on our analyses in two ways. First, we describe a comparison between an Illumina short-read assembly and a PacBio long-read assembly of one of the pan-genome accessions as proposed by the reviewer. Second, we describe how we controlled for potential technical errors and validated our key results using experiments and data independent of genome assemblies. These analyses were reported, but due to space constraints we could not describe them in great detail. We take the opportunity to describe our validation approaches in this response letter more extensively than would be possible in the main text of the manuscript, focussing especially on the potential impact of assembly errors.

1. Comparison of short and long-read assemblies

We followed the reviewer's recommendation to compare long-read and short-read assemblies of one accession to understand the differences between both approaches and their potential impact on our analyses. We sequenced the genome of cv. Morex with PacBio continuous long reads (CLR) and assembled them using MECAT (see Methods, ll. 211-235). This assembly was compared to the short-read assembly, which we used in our pan-genomic analyses (Morex V2, Monat et al. 2019). Morex V2 was constructed with the TRITEX pipeline for the majority of the 20 pan-genome assemblies.

The main results of the comparison are shown in Extended Data Fig. 2 and are summarized here:

- a) As expected, contig N50 is much higher in the long-read assembly (10.2 Mb vs 32.7 kb, Extended Data Fig. 2b). The TRITEX short-read assembly is 5 % longer than the CLR assembly, presumably due to overestimated gap sizes during scaffolding with mate-pair reads (Monat et al. 2019, Genome Biol). Pseudomolecules constructed from the long- and the short-read assemblies show excellent collinearity on a global scale (Extended Data Fig. 2a), as does a local alignment of the *NUDUM* locus, which we use as an example to illustrate the power of GWAS with single-copy pan-genome markers. (Extended Data Fig. 2c).
- b) Gene space completeness is very similar in both assemblies, with 91.0 % aligned full-length cDNAs in the TRITEX and 89.5 % in the CLR assembly. The slightly higher proportion indicate that at least for some genes, the higher accuracy of short-reads may be more beneficial than longer, but error-prone reads.
- c) Aligning a previously published Bionano map (Extended Data Fig. 2b) to both assemblies resulted in slightly more aligned label sites in the CLR assembly (90.5 % [TRITEX] vs. 93.1 % [CLR]). Missing label sites in the short-read assembly are likely due to sequence gaps in repetitive elements.
- d) We used Assemblytics to compare the CLR and TRITEX assembly (Extended Data Fig. 2d). Only few presence-absence variants (PAV) > 1 kb were found between the two assemblies (17 Mb of PAV sequence). The majority of events were differences in the presence or copy number of repetitive elements (253 Mb of sequence). This observation agrees with the expectation that long-read assemblies can better resolve repetitive sequence and vindicates our approach of focussing on single-copy sequences.
- e) We used Assemblytics to detect structural variants between (i) the Morex CLR assembly and the assembly of cv. Barke and (ii) the Morex TRITEX assembly and the Barke assembly. We did find differences in the number and cumulative size of discovered PAVs (Extended Data Fig. 2e). Importantly, however, single-copy regions overlapping PAVs were of very similar size in both assemblies (5.5 +/- 0.1 Mb).

In conclusion, the comparison revealed that both assemblies represent the same genic and single-copy sequence and are highly collinear. As expected a priori and from reports in other species, the long-read assembly resolves repetitive regions of the genome better. We were aware of the potential errors in repetitive regions and hence decided to focus on single-copy regions for quantitative analyses of pan-genome complexity (see also point 2d)

2. Validation analyses

In the following, we describe which approaches we took to ensure that our main results are not affected by technical errors:

a) *2H and 7H inversions*

We analysed in detail two large polymorphic inversions, one closely linked to a flowering time locus (2H inversion) and the other likely the result of mutation breeding (7H inversions). These events were evident both in the Hi-C based pseudomolecules of inversion carriers (RGT Planet [7H] and Barke [2H]) and in a reference-based inversion scan using Hi-C reads mapped to a single sequence assembly (of cv. Morex). We added a figure to the revised manuscript to show the very strong Hi-C signal supporting the 7H inversion in the reference-based analysis (Extended Data Fig. 10d). The genome assemblies of inverted and non-inverted haplotypes were used to design PCR primer pairs spanning the inversion breakpoints. These markers yielded the expected bands (Fig. 3d, Extended Data Fig. 10b,c). Marker data from a population segregating for the 7H locus confirmed the localization of the inversion at the loci predicted by the genome assembly and showed a large region of repressed recombination coincident with inversion boundaries predicted from the RGT Planet genome assembly (Fig. 3b, c). Running the diagnostic marker for the 2H inversion on 200 barley showed (consistent with Hi-C-based inversion calls for 69 barleys) that this event occurs at high frequency in domesticated barley (Supplementary Table 8). SNP genotypes from WGS data shows clear haplotypic differentiation between carriers and non-carriers in SNPs consistent with the absence of meiotic recombination between inverted and non-inverted haplotypes, as expected by theory. In summary, multiple lines of evidence support the presence of these two inversions.

b) *Hi-C based inversion discovery*

We concur with the reviewer that orienting sequence scaffolds with Hi-C data only can be challenging. Our pipeline for Hi-C-based pseudomolecule construction (Monat et al. 2019, Genome Biol) involves automatic and manual steps (including inspection of Hi-C contact matrices) to detect and correct large (> 5 Mb) misassemblies. Smaller inversions introduced either by 10X or Hi-C scaffolding (1-5 Mb) can be hard to detect. The pipeline we used for reference-based inversion discovery (Himmelbach et al. 2018, Plant J) uses the following approach to avoid false positive inversion calls due to mis-oriented sequences: Hi-C data for the reference genotype (Morex in our case) is used as a control. If a mis-oriented scaffold is present, both the test and the control samples will show deviations in the contact matrix. In this case, no inversion is called. In the Himmelbach et al. (2018) paper, we confirmed two large inversions detected by this approach using fluorescence *in situ* hybridization. In conclusion, mis-oriented sequence may be present in our assemblies, but it is unlikely to impact the inversion discovery.

c) *Presence/absence variants*

The present study is the first comprehensive assessment of structural variation at a genome-wide scale in barley. Hence, few genomic datasets of sufficient breadth are available for independent validation. We took a genetic approach to validate structural variants (ll. 148-152). The allelic status of sequences present in the Barke genome, but missing in Morex, was scored in 90 recombinant inbred lines derived from a cross between Morex and Barke. Marker scores were used to place PAVs genetically. Of the 5,602 Barke sequence > 5 kb missing in Morex, 5,446 were mapped genetically to locations concordant with their predicted genomic coordinates (Extended Data Fig. 6d). This analysis provides a validation of larger (> 5 kb) presence-absence events, which can be ascertained with low-depth sequence data of a mapping population.

d) *Single-copy pangenome*

To conduct quantitative estimates of pan-genome complexity (Fig. 2a) and mapping of agronomic traits (Fig. 2b-d, Extended Data Fig. 8), we used what we call “single-copy pangenome markers”, i.e. single-copy sequences showing presence-absence variation and

precisely positioned in chromosome-scale assemblies. We note that single-copy capture also non-repetitive non-genic sequence such as regulatory regions or unique TE insertion sites similar in concept to insertion site-based polymorphisms (ISBPs, Paux et al. 2010 Plant Biotechnol J). We designed this approach not only to circumvent assembly artefacts, but also because of a lack of graph-based pan-genome algorithms capable of handling large eukaryotic genomes (Hickey et al., 2020 Genome Biol).

Again, we took a genetic approach to validate our method. We used WGS data of a global barley diversity panel to genotype single-copy pan-genome markers in 200 domesticated accessions. The resultant marker matrix captured population structure in a meaningful way concordant with SNP data on the same panel (Extended Data Figure 7d-g). Association scans for morphological phenotypes using WGS and GBS data yielded peaks consistent with prior knowledge (Extended Data Fig. 8). Hence, the single-copy pangenome, derived from chromosome-scale assemblies and structural variant calls between them, is a meaningful concept that adds SV to quantitative genetic analysis in the context of plant genetics and breeding.

To summarize, we have shown that differences between long-read and short-assemblies exist mainly in the repetitive intergenic space, but that errors in short-read assemblies have not affected the analyses leading to our main biological conclusions (inversion discovery, single-copy pan-genome).

Finally, we want to briefly explain our rationale for choosing an assembly methodology based on short-reads for the barley pan-genome project. We agree with the reviewer that long-read technology is readily available now. However, at the time we designed the pan-genome project, it was not evident from initial reports about long-read assemblies of Triticeae species (Zimin et al. 2017 Genome Res, Zimin et al. 2017 GigaScience) whether this approach would scale to chromosome-scale assembly of 10-20 accessions, because the reported contiguity for diploid wild wheat (*Ae. tauschii*) was below 1 Mb (N50: 486 kb) and compute time to obtain a primary assembly was on the order of several months. By contrast, the short-read assemblies underlying the current reference genome sequences of barley (Monat et al. 2019 Genome Biol) and wheat (IWGSC 2018, Science) were assembled from short-reads in the time frame of 4-6 weeks and had N50 values > 2 Mb.

Only very recent advances in long-read sequencing and assembly methodology made it possible to tackle the very large and repeat-rich genomes of barley and wheat to enable chromosome-scale assemblies of multiple genotypes. Data collection for the Morex CLR assembly reported here was initiated only when all pan-genome had been completed and primary contig assembly still required several months. We expect that a novel method to obtain accurate long-reads (PacBio HiFi, Wenger et al. 2019, Nat Biotech) will further improve and accelerate long-read assembly, and we will explore this approach for future pan-genomic studies in barley

Reviewer Reports on the First Revision:

Referee #2 (Remarks to the Author):

General comment: This is a review of a revised version. Please see my comments on A-H in my original review.

Specific comments:

The authors have addressed all my points raised on the original submission. They have clarified the origin of the wild barley accession used and referred to earlier (non-genomic) data on this genotype.

Furthermore, they provide additional data and analysis on gene counts (Extended data Fig. 5e) and provide an expanded analysis of the NUD locus, both in response to my comments.

Finally, Reviewer 3 has raised a similar point as I did in my review on structural variation and asked for long read analysis. Obviously, this analysis also responds to my comment, in much more detail than I was requesting.

Referee #3 (Remarks to the Author):

I'm quite happy with the paper's efforts to verify results with exemplar long read genomes. So long as these long reads and/or long read assemblies (and descriptions thereof) are included in the publication, I think the paper is suitable now for acceptance after meeting other reviewers' concerns.